# Methylation of lysine 36 on histone H3 is required to control transposon activities in somatic cells

Henrik Lindehell, Yuri B Schwartz, Jan Larsson

**Transposable elements constitute a substantial portion of most eukaryotic genomes and their activity can lead to developmental and neuronal defects. In the germline, transposon activity is antagonized by the PIWI-interacting RNA pathway tasked with repression of transposon transcription and degrading transcripts that have already been produced. However, most of the genes required for transposon control are not expressed outside the germline, prompting the question: what causes deleterious transposons activity in the soma and how is it managed? Here, we show that disruptions of the Histone 3 lysine 36 methylation machinery led to increased transposon transcription in *Drosophila melanogaster* brains and that there is division of labour for the repression of transposable elements between the different methyltransferases Set2, NSD, and Ash1. Furthermore, we show that disruption of methylation leads to somatic activation of key genes in the PIWI-interacting RNA pathway and the preferential production of RNA from dual-strand piRNA clusters.**

## Introduction

Transposable elements make up a large part of the eukaryotic genome and come in a multitude of variants. They have been shown to drive gene regulation and to exhibit sophisticated ways of selective insertion in the genome (Bourque et al, 2018), thereby making transposon activity a fundamental driver of evolution and gene regulation (Moran et al, 1999; Koga, 2006; Senft & Macfarlan, 2021). The processes leading to transposon activity have traditionally been studied in the germline because only these cells can pass their modified genomes to the offspring. However, a series of recent studies put the spotlight on transposon activity in somatic cells by linking it to disease, cellular ageing, cancer, and developmental processes, for example, brain development (Faulkner & Garcia-Perez, 2017; Loreto & Pereira, 2017; Evans & Erwin, 2021; Tsuji et al, 2021). To repress transcription of transposable elements, organisms have evolved adaptive surveillance and interference systems consisting of short noncoding PIWI-interacting RNA (piRNA)-binding Argonaut

class proteins to cleave transposable element transcripts in a sequence-specific manner (Czech & Hannon, 2016). The piRNAs make up the single largest group of noncoding RNAs and differs from other small noncoding RNAs in length, complexity, and strand specificity (Siomi et al, 2011). The PIWI/piRNA biosynthesis process is further fine-tuned by the incorporation of Tudor domain containing proteins such as Krimper which assembles the ping-pong amplification complex for more efficient targeting of transposable element transcripts (Webster et al, 2015). The PIWI/piRNA biosynthesis proteins are in some instances expressed in somatic cells, the function of which is still being debated (Cheng et al, 2014; Tosar et al, 2018; Galton et al, 2022). In addition, DNA methylation (in the form of 5-methyl-cytosine) is prevalent in eukaryote genomes and appears to be important for transcriptional silencing of transposable elements in somatic cells (Greenberg & Bourc'his, 2019; Yoder et al, 1997).

Di- or trimethylation of lysine 36 on histone H3 (H3K36me2/3) has been extensively studied and is foremost associated with transcriptional activity. However, the role of H3K36 methylation is complex and it has been shown to be a key player in preventing run-away cryptic transcription (Li et al, 2009a), DNA repair (Fnu et al, 2011), and chromosome-specific gene regulation (Lindehell et al, 2021). Disruption of H3K36 methylation has been linked to various cancers and cognitive diseases (Kurotaki et al, 2002; Newbold & Mokbel, 2010; Xiao et al, 2021) but also to the maintenance of genomic DNA methylation patterns. Thus, in mammalian cells, histone H3K36me2 was shown to facilitate the binding of the DNA methyltransferase DNMT3A to chromatin and thereby promote DNA methylation in intergenic regions. In *Drosophila melanogaster*, three evolutionary conserved proteins methylate H3K36. SET domain containing 2 (Set2) is the only histone methyltransferase to trimethylate H3K36 and *Set2* mutants show a 10-fold reduction of H3K36me3, whereas bulk mono- and dimethylated H3K36 remain largely unaffected (Larschan et al, 2007; Dorafshan et al, 2019). Absent, small or homeotic discs 1 (Ash1) can add either one or two methyl groups to H3K36 and loss-of-function in Ash1 results in a twofold reduction in H3K36me1 but no detectable loss of bulk di- and trimethylated H3K36 (Dorafshan et al, 2019). Less is known about the substrates of *Drosophila* Nuclear receptor-binding SET domain protein (NSD) but closely related mammalian orthologs (NSD1, NSD2, and NSD3) have been shown to mono and dimethylate

Department of Molecular Biology, Umeå University, Umeå, Sweden

Correspondence: jan.larsson@molbiol.umu.se

H3K36 (Rayasam, 2003; Li et al, 2009b). *NSD* loss-of-function mutants show no obvious changes in levels of bulk mono-, di- or trimethylated H3K36 (Dorafshan et al, 2019).

Mutations causing substitution of lysine 36 to methionine act in a dominant negative fashion by inhibiting the activity of the H3K36-specific methyltransferases (Lewis et al, 2013). Somatic H3K36M mutations are common in several cancers (Lewis et al, 2013; Lu et al, 2016; Papillon-Cavanagh et al, 2017). Intriguingly, recent work has shown that overexpression of H3.3K36M in eye imaginal discs of *D. melanogaster* larvae causes increased transcription of transposable elements (Chaouch et al, 2021). It would have been tempting to speculate that the observed effect is caused by disrupted H3K36 methylation and subsequent changes in DNA methylation landscape. However, *D. melanogaster* lacks DNA methylation and an alternative explanation is therefore required. The explanation is likely to be complicated. The overexpression of H3.3K36M acts by inhibiting H3K36-specific methyltransferases. It remains unknown through which methyltransferase the activation of transposable element is mediated, if this is via disrupted H3K36 methylation and if so, through which methylation state of H3K36. These are central questions because there is evidence that at least two of the three *Drosophila* H3K36 methyltransferases (Set2 and Ash1) also target, yet unknown, non-histone proteins. The loss of Set2 impairs dosage compensation of the male X chromosome independent of H3K36 methylation (Lindehell et al, 2021) and the role of Ash1 in preventing excessive Polycomb repression may be independent of its role as a H3K36 methyltransferase (Dorafshan et al, 2019).

It has been demonstrated that in *D. melanogaster*, the expression of retrotransposons increases with age (Li et al, 2013). The most plausible explanation is the ageing-related decline of trimethylation of histone H3 at lysine-9 (H3K9me3) and heterochromatin protein 1 in pericentromeric regions where most transposons are located (Wood et al, 2010; Bulut-Karslioglu et al, 2014). However, in *D. melanogaster*, H3K36 methylation also declines with age (Wood et al, 2010), and considering the direct link between H3K36 methylation and the subsequent recruitment of DNA methyltransferases in mammals (Weinberg et al, 2019; Xu et al, 2020) together with the transcriptional activation of transposons in H3K36M replacement mutant flies (Chaouch et al, 2021), it is crucial to study the role of H3K36 methylation as a repressor of transposable-element activity. Because *D. melanogaster* lacks DNA methylation, the role of H3K36 methylation in transposon silencing is intriguing from a functional and an evolutionary perspective with a potentially ancient role of H3K36 methylation as a defence against foreign DNA insertion predating its link to DNA methylation.

To investigate and dissect the role of H3K36 methylation and associated histone methyltransferases in control of transposable-element activity, we analysed the transcript levels of transposable elements in dissected brains from third instar male larvae mutants of the three known methyltransferases: Set2, NSD, and Ash1 and the replacement of histone H3 with a variant in which lysine 36 is replaced with arginine, H3K36R. Third instar larvae stage was chosen as it is the last viable stage for all mutant strains and brain was chosen because it is a diploid tissue with minor maternal contribution. Males were used to address questions about possible effects of dosage compensation of the single male X (Lindehell et al, 2021).

We show here that H3K36 methylation of the replication-dependent Histone 3 represses transcription of transposable elements. The three methyltransferases display the division of labour, and the main effect is mediated via Set2 and most likely through H3K36me2. Furthermore, the increased transcription of transposable elements induces transcriptional activation of genes encoding proteins that comprise the PIWI surveillance system and production of RNAs from piRNA clusters, preferentially from dual-strand piRNA clusters, in somatic cells.

# Results

## Transcriptional activation of transposable elements in the absence of K36 methylation on replication-coupled histone H3

To test the potential effect of lost methylation on the replication-dependent H3K36 (H3.2K36) on transposon repression, we analysed transcript levels of transposable elements in dissected brains from third instar male larvae with the lysine-to-arginine replacement mutant Δ*HisC*; *12x^{H3K36R}* (Günesdogan et al, 2010; McKay et al, 2015). We have previously shown that trimethylated H3K36 is strongly reduced in brain and imaginal discs from mutant Δ*HisC*; *12x^{H3K36R}* larvae (Lindehell et al, 2021). The Δ*HisC*; *12x^{H3K36R}* mutant does not share the methyltransferase toxicity observed upon overexpression of H3K36M (Lu et al, 2016). Nevertheless, we observed a significant increased transcript abundance of transposable elements in Δ*HisC*; *12x^{H3K36R}* compared with control, Δ*HisC*; *12x^{H3K36K}* (Fig 1A). In our analysis, we measure increased transcript abundance which we assume is mainly caused by increased transcription. The transcriptionally activated transposons in Δ*HisC*; *12x^{H3K36R}* share roughly the same superfamily ratios as upon overexpression of H3.3K36M where most of the transcribed transposons are LTR of Gypsy, Pao, and Copia family and Jockey family LINE repeats (Figs 1B and S1) (Chaouch et al, 2021). These transposon families are being highly favoured when compared with the expected ratios between different elements (Figs 1B and S1). We next analysed the genomic locations of the 4,088 significantly up-regulated transposons (log$_2$FC > 2, *P* < 0.05) and found that the bulk of the transcriptional activation takes place in the pericentromeric regions of all chromosomes and the entire sequenced part of the Y chromosome (Fig 1C). To verify that the increase in transposable elements is not the result of increased transcription of genes harbouring transposable elements, we quantified LTR and LINE elements within active and inactive genes and within intergenic regions. We found that most (71.7%) of up-regulated transposons in Δ*HisC*; *12x^{H3K36R}* is associated with intergenic regions and only 7.3% is associated with genes transcribed in the wild type animals (Fig 1D). Genes that are silent in the wild type animals accommodate 21% of up-regulated LTR and LINE elements in Δ*HisC*; *12x^{H3K36R}* and within this group only 10.5% of the transposons localize to genes that are transcriptionally activated in Δ*HisC*; *12x^{H3K36R}* (Fig 1D). Therefore, we conclude that an increased transcription of genes' harbouring transposons is not responsible for the dramatic increase in transposon transcription in Δ*HisC*; *12x^{H3K36R}*. To factor in the importance of the replication-independent H3.3 variant histone, we analysed the complete H3.3 knock-out mutant Δ*H3.3B*; Δ*H3.3A* and found no significant transcriptional activation of transposable elements (Fig 1E). We conclude that keeping transposable elements transcriptionally inactive requires uncompromised replication-dependent H3K36.

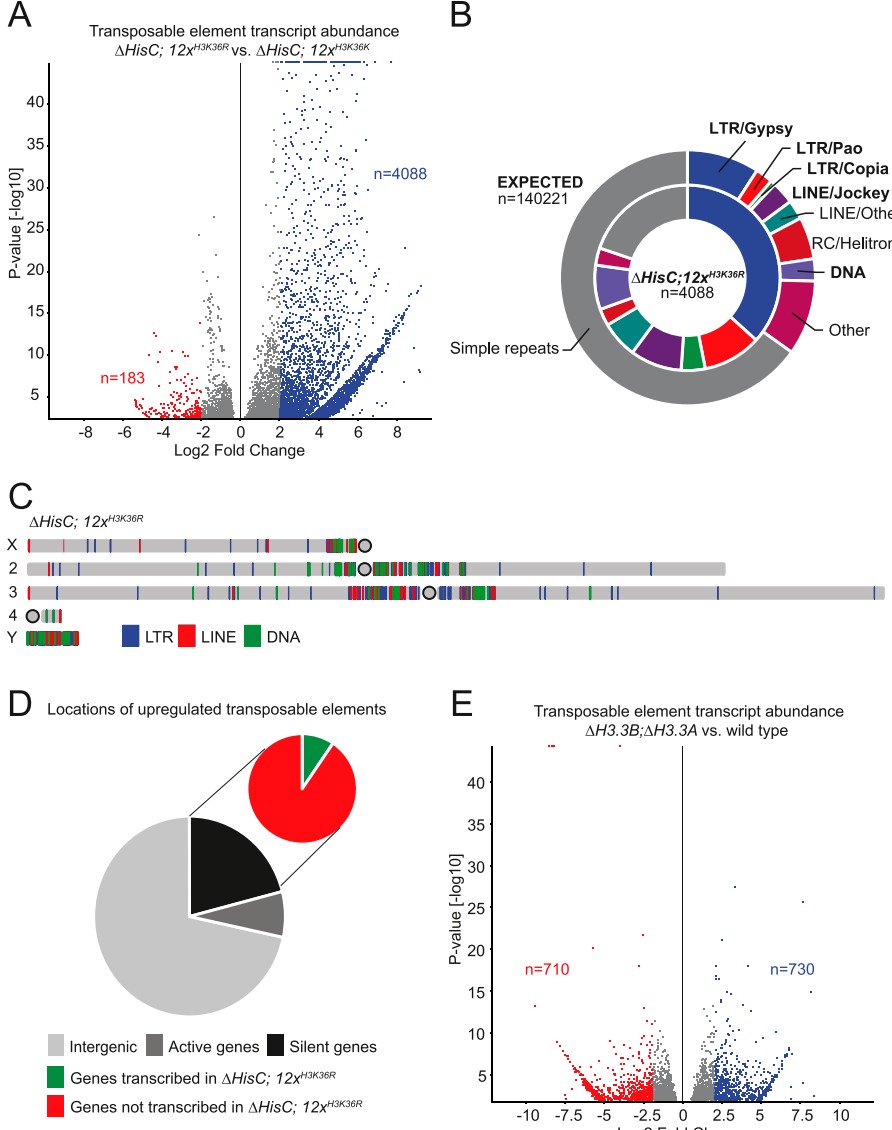

**Figure 1. Loss of H3K36 methylation triggers transcription of transposable elements in somatic cells.**

**(A)** Volcano plot showing differential transcription of transposable elements between $\Delta HisC$; $12x^{H3K36R}$ and control, $\Delta HisC$; $12x^{H3K36K}$. Shown in blue are 4,088 transposons up-regulated by $\log_2$ fold change >2 ($P < 0.05$). Note that the y-axis in this and all following volcano plots starts at 1.3, corresponding to $P = 0.05$. **(B)** Pie chart representing the expected ratio between transposon families based on genome abundance (outer circle) versus the observed transcription of 4,088 transposons in $\Delta HisC$; $12x^{H3K36R}$ (inner circle). **(C)** Genomic locations of transposable elements up-regulated by $\log_2$ fold change >2 ($P < 0.05$) in $\Delta HisC$; $12x^{H3K36R}$. **(D)** The large pie chart shows the proportional localization of up-regulated LTR and LINE elements within intergenic regions (light grey), wild type expressed genes (medium grey), and wild type unexpressed genes (black). The small pie chart shows the division between up-regulated LTR and LINE elements within transcriptionally activated genes (green) and genes that remain unexpressed in $\Delta HisC$; $12x^{H3K36R}$ (red). **(E)** Volcano plot showing differential transcription of transposable elements between $\Delta H3.3B$; $\Delta H3.3A$, and wild type.

## Division of labour and differential transcriptional activation of transposable elements in histone methyltransferase mutants

The absence of increased transposon activities in $\Delta H3.3B$; $\Delta H3.3A$ together with the methyltransferase inhibitory effects upon over-expression of H3.3K36M prompted us to further investigate the division of labour between the three known H3K36 methyl-transferases; Set2, NSD, and Ash1. By analysing RNA sequencing data from *Set2*, *ash1*, and *NSD* knock-out mutants, we found that loss of Set2 led to transcriptional activation of transposable elements comparable in degree to $\Delta HisC$; $12x^{H3K36R}$ (n = 3,057, $\log_2$FC > 2, $P < 0.05$) (Fig 2A). $NSD^{ds46}$ also showed increased transcriptional activation (n = 998, $\log_2$FC > 2, $P < 0.05$) (Fig 2B). The $ash1^{22}/ash1^{9011}$ mutant showed the lowest levels of transposons being activated (n = 685, $\log_2$FC > 2, $P < 0.05$) (Fig 2C). To better understand the contribution of each methyltransferase, we ran a cluster analysis

for $Set2^1$, $NSD^{ds46}$, and $ash1^{22}/ash1^{9011}$ on repeats whose transcription increased more than twofold in $\Delta HisC$; $12x^{H3K36R}$. The set of repeated elements that become transcribed after the loss of Set2 represents most of the transposons activated in $\Delta HisC$; $12x^{H3K36R}$. The remaining repeats are included in sets that become more transcriptionally active in $NSD^{ds46}$ and $ash1^{22}/ash1^{9011}$ mutants. These two groups of repeats are largely distinct from each other and from those affected by the Set2 loss, indicating division of labour between the H3K36 histone methyltransferases (Fig 2D). Analysing the histone–methyltransferase-specific clusters (Fig 2D) compared with theoretically unbiased transcriptional activation of transposons, we found that preferential transcriptional activation of LTR transposons is a feature of all histone methyl-transferase mutant strains. In the $Set2^1$-specific cluster, 65.4% of all transposons being activated are of the LTR superfamily, in $NSD^{ds46}$ 29.9% are LTRs and in $ash1^{22}/ash1^{9011}$ 23.3% belong to the

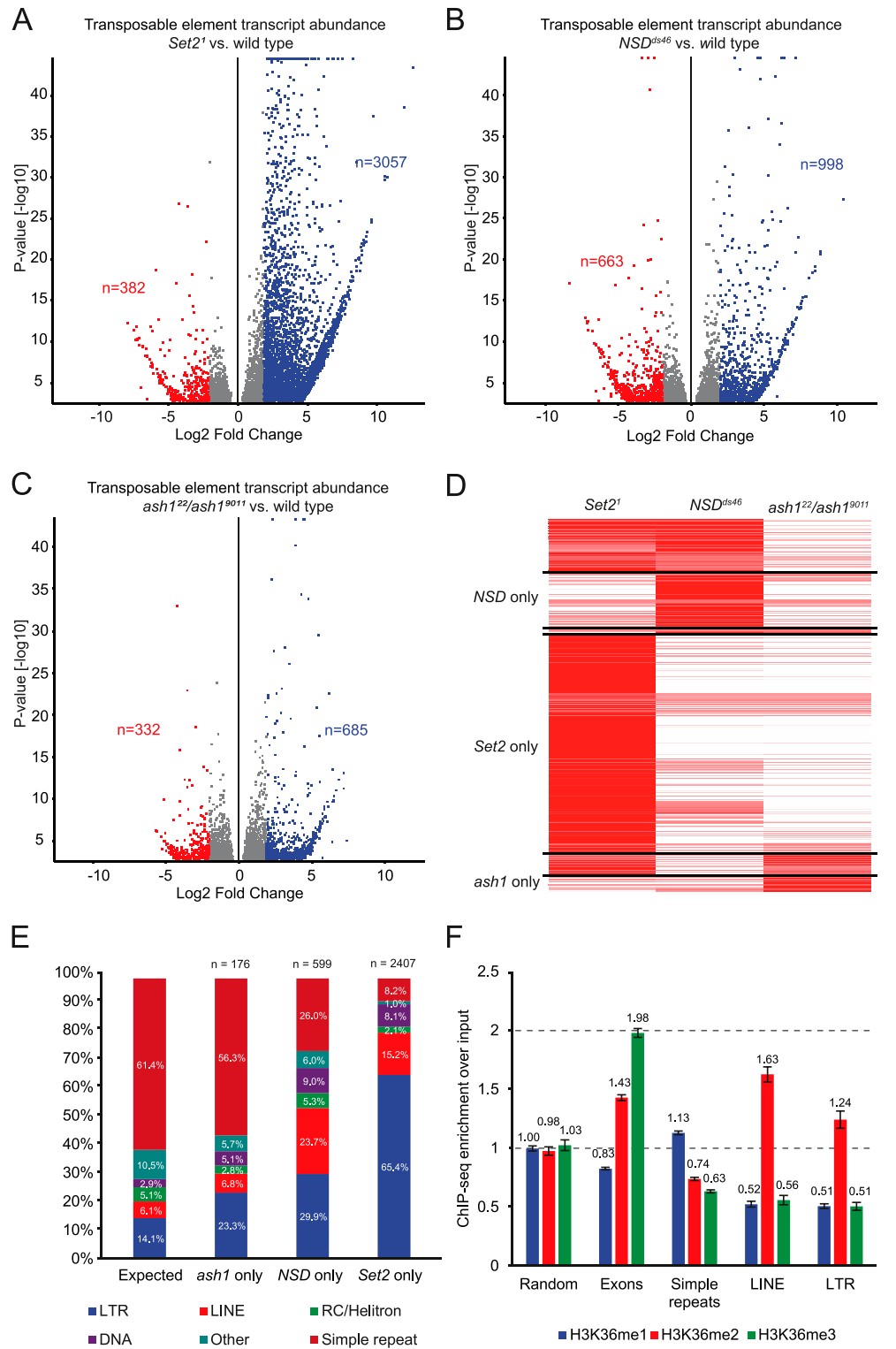

**Figure 2. Division of labour for transposon repression between the three H3K36 methyl transferases.**
**(A)** Volcano plot showing differential transcription of transposable elements between $Set2^1$ and wild type. Shown in blue are 3,057 transposons being transcriptionally activated by $\log_2$ fold change >2 ($P < 0.05$). **(B)** Volcano plot showing differential transcription of transposable elements between $NSD^{ds46}$ and wild type. Shown in blue are 998 transposons being up-regulated by $\log_2$ fold change >2 ($P < 0.05$) **(C)** Volcano plot showing differential expression of transposable elements between $ash1^{22}$/$ash1^{9011}$ and wild type. Shown in blue are 685 transposons differentially transcribed by $\log_2$ fold change >2 ($P < 0.05$) **(D)** Heatmap comprised of transposons transcriptionally activated in $\Delta HisC$; $12X^{H3K36R}$ and their level of transcription in the corresponding methyltransferase mutants: $Set2^1$, $NSD^{ds46}$ and $ash1^{9011}$. n(Set2) = 2,407, n(Set2 + NSD) = 595, n(Set2 + ash1) = 242, n(NSD) = 599, n(NSD + ash1) = 60, n(ash1) = 176. **(E)** Transcriptional distribution between different orders of transposons in $Set2^1$, $NSD^{ds46}$, and $ash1^{22}$/$ash1^{9011}$ exclusive clusters (See (D) above) compared with unbiased expected ratios based on genome abundance. **(F)** ChIP-seq over input ratios for H3K36me1 (blue), H3K36me2 (red), H3K36me3 (green) in randomly sampled positions (n = 10,000), exons (n = 15,340), simple repeats (n = 85,899), LINE repeats (n = 6,988), and LTR repeats (n = 15,889). Error bars represent 95% confidence interval.

LTR order (Fig 2E and Table S1). Noteworthy, 23.7% of repeats transcribed in $NSD^{ds46}$ mutants belong to the LINE superfamily, which is four times higher than expected from a hypothetical unbiased transcriptional activation (Fig 2E and Table S1). If activation of transposons would be random, based on genome abundance, we would expect 61.4% of the repeat RNA to originate from simple repeats (duplications of simple sets of DNA bases) (Fig 2E). Interestingly, we find that the fraction of simple repeats in

$ash1^{22}/ash1^{9011}$, $NSD^{ds46}$, and $Set2^1$ is 56.3%, 26.0%, and 8.2%, respectively (Fig 2E and Table S1). Chi-square tests between expected and observed for all conditions and between $NSD^{ds46}$ and $Set2^1$ were all $P < 0.001$ and therefore the differences in distribution cannot be explained by random activation. We observe no clear bias for genomic locations of transposons transcribed in $Set2^1$, $NSD^{ds46}$, and $ash1^{22}/ash1^{9011}$. All histone methyltransferases do, to a varying degree, repress repeat transcription around autosomal and X-chromosome centromeres and the entire sequenced region of the Y chromosome (Fig S2).

The increased transposon transcription observed in Δ$HisC$; $12x^{H3K36R}$ is to a large extent recapitulated in $Set2^1$, indicating that disrupted methylation of H3K36 causes increased transposon transcription. We next asked if this activation can be linked to one of the three methylation states. To this end, we analysed ChIP-seq data for mono-, di-, and trimethylated H3K36 from adult fly heads to test if different transposons vary in their H3K36 methylation levels. We observe a significant difference between LINE and LTR compared with simple repeats or genome average where both LINE and LTR transposons have higher relative levels of H3K36me2 but lower levels of H3K36me1 and H3K36me3, whereas simple repeats show a slight but significant elevation of the more ubiquitous H3K36me1 (Fig 2F). The LINE and LTR transposons also show H3K36 methylation profiles distinct from exons and randomly sampled regions (Fig 2F). We conclude that all three H3K36 histone methyltransferases, presumably through the dimethylation of H3K36, help to prevent transcription of transposable elements and that Set2 is the main methyltransferase responsible for this.

### Loss of H3K36 methylation triggers transcription within dual-strand piRNA clusters

Because repeated elements are being transcribed, we asked if we can detect transcription within piRNA clusters, the prerequisite for the piRNA production. Earlier studies have shown increase in piRNA production upon overexpression of H3.3K27M and to a lesser degree also after overexpression of H3.3K36M (Chaouch et al, 2021). By mapping total RNA-seq reads to annotated piRNA clusters, we found significant increase in putative precursor piRNAs in Δ$HisC$; $12x^{H3K36R}$ (n = 3,886, $\log_2$FC > 2, $P < 0.05$) (Fig 3A) and in $Set2^1$ (n = 2,678, $\log_2$FC > 2, $P < 0.05$) (Fig 3B). Similar to what is observed with transposon transcription, $NSD$ mutants also produce more putative piRNA precursors (n = 2,319, $\log_2$FC > 2, $P < 0.05$) (Fig 3C). In $ash1$ mutants, we observe a more modest transcription across piRNA clusters (n = 680, $\log_2$FC > 2, $P < 0.05$) (Fig 3D), which suggests that the loss of H3K36 methylation mediated via Set2 and NSD are the most significant triggers of transcription within these loci. By quantifying normalized number of reads per annotated piRNA cluster, we found that the piRNA cluster transcripts were most abundant in the Δ$HisC$; $12x^{H3K36R}$ mutants with an average of 24.4 reads specifically mapped to individual precursor templates (Fig 3E). Corresponding numbers for methyltransferase mutants and wild-type control were as follows: $Set2^1$ = 14.5, $NSD^{ds46}$ = 9.5, $ash1^{22}/ash1^{9011}$ 6.4, and wild type = 5.1 (Fig 3E). Overall, the degree of transcription within piRNA clusters in different mutants correlates to the levels of transposon transcription. This suggests that increased transcription within piRNA clusters is a consequence of the increased transcription of

repeated elements rather than the loss of H3K36 methylation from the piRNA clusters.

The piRNA clusters in *D. melanogaster* are defined as two general types based on their mode of transcription in the germline (Brennecke et al, 2007; Mohn et al, 2014). Uni-strand clusters are transcribed on one strand from distinct RNA polymerase II promoters (Brennecke et al, 2007; Goriaux et al, 2014; Mohn et al, 2014). In contrast, transcription of dual-strand piRNA clusters embedded in H3K9me3-enriched heterochromatin depends on the presence of Rhino, a germline-specific heterochromatin protein 1 paralog specifically enriched in these clusters (Klattenhoff et al, 2009; Vermaak & Malik, 2009). The dual-strand clusters lack defined promoters and H3K9me3 and Rhino are required to recruit the transcriptional machinery. To evaluate whether one of the two mechanisms is preferentially triggered by the loss of H3K36 methylation, we compared uni-strand and dual-strand clusters. Clusters *42AB* and *38C* (Brennecke et al, 2007) are both dual-strand and dependant on the Rhino pathway for their transcription, whereas cluster *20A* and the *flamenco* cluster represent uni-strand transcribed piRNA clusters. We analysed the mean read count mapped to precursor templates within individual piRNA clusters and found the dual-strand piRNA clusters to be more transcriptionally active than uni-strand clusters. For the dual-strand *42AB* cluster, we measured a 12.7-fold increase in mapped RNA-seq reads in Δ$HisC$; $12x^{H3K36R}$ compared with wild type (Fig 3F). The fold increase in RNA-seq reads over the wild type for $Set2^1$, $NSD^{ds46}$, and $ash1^{22}/ash1^{9011}$ are 7.0, 4.3, and 2.2, respectively (Fig 3F). For the dual-strand cluster *38C*, the corresponding numbers for Δ$HisC$; $12x^{H3K36R}$, $Set2^1$, $NSD^{ds46}$, and $ash1^{22}/ash1^{9011}$ are: 7.1, 5.7, 2.6, and 1.5 (Fig 3F). For the uni-strand clusters, the difference in the RNA-seq reads mapped to piRNA clusters is subtle: 1.7, 1.3, 1.2, 1.4 for cluster *20A* and 3.1, 1.9, 1.9, 2.2 for the *flamenco* cluster (Fig 3F). The loss of H3K36 methylation through Δ$HisC$; $12x^{H3K36R}$ has the most pronounced effect on dual-strand piRNA cluster activity. We also observe that the different methyltransferases contribute to different degrees, mirroring the effect size observed for transcriptional activation of transposable elements (Fig 2A–C). The largest effect on dual-strand clusters is observed in $Set2^1$, followed by $NSD^{ds46}$ and $ash1^{22}/ash1^{9011}$ in descending order. For uni-strand clusters, we observe only a small increase in RNA-seq signal within piRNA clusters which is equally associated with each of the H3K36 methyltransferases.

### The piRNA biosynthesis pathway is activated upon compromised H3K36 methylation

We were intrigued by the apparent increased transcription within the dual-strand clusters in Δ$HisC$; $12x^{H3K36R}$ because these clusters have been shown to require PIWI machinery components such as Rhino, Moonshiner, and Deadlock for their transcription (Mohn et al, 2014; Andersen et al, 2017). This motivated us to analyse if genes encoding PIWI machinery proteins, including Argonate family proteins, Tudor domain proteins, and other nuage proteins, display higher transcription in mutants with impaired H3K36 methylation. In Δ$HisC$; $12x^{H3K36R}$, we observe transcriptional activation of almost all genes linked to piRNA biogenesis, including all three Argonaut protein encoding genes: *AGO3*, *aub*, and *piwi* (Table 1). Among the

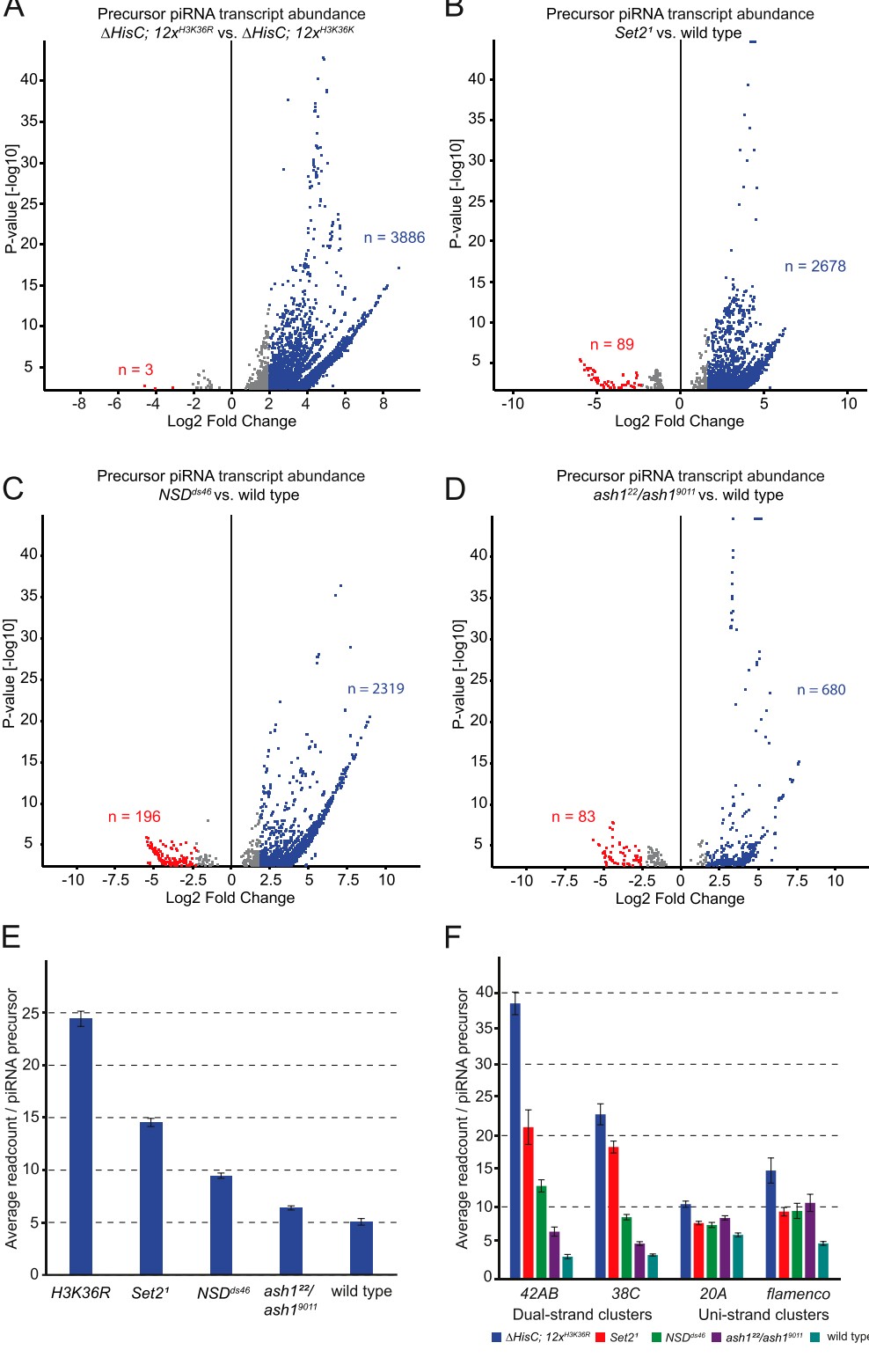

**Figure 3. Dual-strand piRNA clusters are activated in the absence of H3K36 methylation.**
**(A)** Volcano plot showing differential transcription of piRNA precursors between ΔHisC; 12x$^{H3K36R}$ and ΔHisC; 12x$^{H3K36K}$. Shown in blue are 3,886 piRNA precursors being up-regulated by log$_2$ fold change >2 ($P < 0.05$). **(B)** Volcano plot showing differential transcription of piRNA precursors between Set2$^1$ and wild type. Shown in blue are 2,678 piRNA precursors being up-regulated by log$_2$ fold change >2 ($P < 0.05$). **(C)** Volcano plot showing differential transcription of piRNA precursors between NSD$^{ds46}$ and wild type. Shown in blue are 2,319 piRNA precursors being up-regulated by log$_2$ fold change >2 ($P < 0.05$). **(D)** Volcano plot showing differential transcription of piRNA precursors between ash1$^{22}$/ash1$^{9011}$ and wild type. Shown in blue are 680 piRNA precursors being up-regulated by log$_2$ fold change >2 ($P < 0.05$). **(E)** Bar chart showing the average number of reads mapping to precursor piRNAs originating from annotated piRNA clusters for different H3K36 methylation mutant conditions. n(H3K36R) = 24.44, n(Set2) = 14.54, n(NSD) = 9.47, n(ash1) = 6.38, n(WT) = 5.06. **(F)** Bar chart showing average number of reads mapping to dual-strand piRNA clusters 42AB and 38C (left) and uni-strand clusters 20A and flamenco (right). The cluster average read number per piRNA precursor in ΔHisC; 12x$^{H3K36R}$ (blue), Set2$^1$ (red), NSD$^{ds46}$ (green), ash1$^{22}$/ash1$^{9011}$ (purple), and wild type (teal), respectively: 38.51, 21.16, 12.93, 6.53, and 3.03 for cluster 42AB; 22.96, 18.38, 8.55, 4.85, and 3.25 for cluster 38C; 10.37, 7.72, 7.45, 8.45, and 6.06 for cluster 20A; 15.09, 9.32, 9.43, 10.56, and 4.88 for the flamenco cluster. Error bars in (E, F) represent 95% confidence interval.

significantly up-regulated genes were also *rhino*, *deadlock*, and *moonshiner* which are directly involved in dual-strand piRNA cluster transcription (Andersen et al, 2017). We note that *cutoff* (*cuff*), proposed to aid the transcription of piRNA source loci via read-through from flanking genes (Mohn et al, 2014; Chen et al, 2016; Andersen et al, 2017), is not increased in the transcript level. However, the gene is transcribed in both ΔHisC; 12x$^{H3K36R}$ and ΔHisC; 12x$^{H3K36K}$. Gene ontology for the top 500 most up-regulated genes

**Table 1. PIWI/piRNA biosynthesis genes are transcribed in the absence of H3K36 methylation.**

| | Gene | H3K36R log₂FC | P-value | Set2[KO] log₂FC | P-value | NSD[KO] log₂FC | P-value | ash1[KO] log₂FC | P-value |
|---|---|---|---|---|---|---|---|---|---|
| Argonaut family | *AGO3* | 2.24 | **** | 0.66 | *** | −2.04 | **** | −0.92 | **** |
| | *piwi* | 1.45 | **** | 0.27 | ns | −0.42 | ns | −0.5 | * |
| | *aub* | 0.82 | ** | 0.74 | *** | −0.29 | ns | 0.21 | ns |
| Heterochromatin piRNA transcription | *moon* | 2.45 | **** | 0.34 | ns | 0.58 | * | 0.08 | ns |
| | *del* | 1.08 | **** | −1.63 | **** | −0.54 | **** | −0.91 | **** |
| Nuclear proteins involved in piRNA biogenesis | *rhi* | 0.97 | **** | −0.12 | ns | 0.48 | ns | 0.59 | * |
| | *Hel25E* | 0.03 | ns | 0.04 | ns | −0.11 | ns | −0.16 | * |
| | *cuff* | 0.027 | ns | −0.47 | * | −0.84 | **** | −0.72 | *** |
| Tudor domain proteins | *CG15930* | 5.5 | **** | 0.92 | **** | 0.02 | ns | −0.23 | ns |
| | *Kots* | 3.96 | **** | 0.06 | ns | −0.12 | ns | 0.08 | ns |
| | *krimp* | 3.91 | **** | −0.13 | ns | −1.69 | **** | −1.19 | **** |
| | *fs(1)Yb* | 2.37 | **** | 1.3 | **** | 0.45 | ns | −0.13 | ns |
| | *qin* | 1.94 | **** | 0.48 | ns | 0.2 | ns | 0.62 | * |
| | *BoYb* | 1.52 | **** | 0.74 | ** | −0.32 | ns | −0.17 | ns |
| | *tej* | 0.74 | * | −0.52 | * | 0.44 | ns | 0.21 | ns |
| | *SoYb* | 0.63 | ** | −1.43 | **** | −1.2 | **** | −0.21 | ns |
| | *vret* | 0.42 | * | −0.05 | ns | −0.61 | **** | −0.12 | ns |
| | *spn-E* | 0.37 | ** | −0.25 | ns | −0.2 | ns | −0.73 | **** |
| | *egg* | 0.12 | ns | 0.02 | ns | 0.03 | ns | −0.01 | ns |
| | *tud* | 0.02 | ns | 0.03 | ns | −0.1 | ns | 0.06 | ns |
| | *papi* | −0.01 | ns | −0.02 | ns | 0.16 | ns | 0.23 | ns |
| Other nuage proteins | *vas* | 1.79 | **** | −1.28 | **** | −0.29 | ns | −1.11 | **** |
| | *shu* | 0.57 | ** | −1.24 | **** | −0.73 | **** | −0.43 | * |
| | *zuc* | 0.047 | ns | −0.7 | ** | −0.28 | ns | −0.12 | ns |
| | *armi* | 0.15 | ns | −0.31 | **** | −0.2 | * | −0.05 | ns |
| | *squ* | 0.08 | ns | −0.02 | ns | 0.02 | ns | 0 | ns |
| | *mael* | −0.73 | ** | 0.13 | ns | 0.65 | ** | 0.87 | *** |

Table showing log₂ fold change values for differentially transcribed genes related to the PIWI/piRNA biosynthesis machinery. ns = $P > 0.05$, * = $P \leq 0.05$, ** = $P \leq 0.01$, *** = $P \leq 0.001$, **** = $P \leq 0.0001$.

($P < 0.05$) in Δ*HisC*; *12x$^{H3K36R}$* showed significant enrichment for genes related to piRNA metabolic processes (Table S2). For *Set2$^1$*, we observe significant up-regulation of *AGO3* and *aub* together with genes encoding the Tudor domain proteins, *fs(1)Yb* and *BoYb* (Table 1). We note a 5.5 log₂ fold up-regulation of *CG15930* (Tudor domain containing protein 5-like) in Δ*HisC*; *12x$^{H3K36R}$*. *CG15930* has been implicated in the piRNA metabolic process and promoting male sexual identity in the germline (Primus et al, 2019). Significant up-regulation of *CG15930* is also found in the *Set2$^1$* samples. The function of *CG15930* in somatic cells and its link to the piRNA machinery remains an open question which we have not pursued further. In Δ*HisC*; *12x$^{H3K36R}$* we also observe a 4.0 log₂ fold up-regulation of the gene *Kotsubu* (*Kots*). Kots is a germline-specific Tudor domain containing protein essential for maintaining high piRNA levels produced from uni-strand and dual-strand piRNA clusters (Lim et al, 2022). In *NSD$^{ds46}$* and *ash1$^{22}$/ash1$^{9011}$*, there is no significant increase in transcription of genes encoding

Argonaut proteins and only a handful of the remaining piRNA biosynthesis genes are being transcribed (Table 1).

**Impaired H3K36 methylation increases transcription of repeated elements and components of the piRNA pathway in other larval tissues**

Finally, we asked if the observed effects on transposon activity, piRNA biosynthesis, and dual-strand cluster activation also occurred outside of eye discs and larval brains. To this end, we reanalysed the results of sequencing nuclear RNA from Δ*HisC*; *12x$^{H3K36R}$* and Δ*HisC*; *12x$^{H3K36K}$* whole larvae of both sexes (Meers et al, 2017). We observed increased transposon transcription in both sexes (Fig 4A and B) with similar relative abundances between different classes of transposons as observed in the brains and in the eye discs (Figs 1B, 4C and D, and S1). We also observed increased

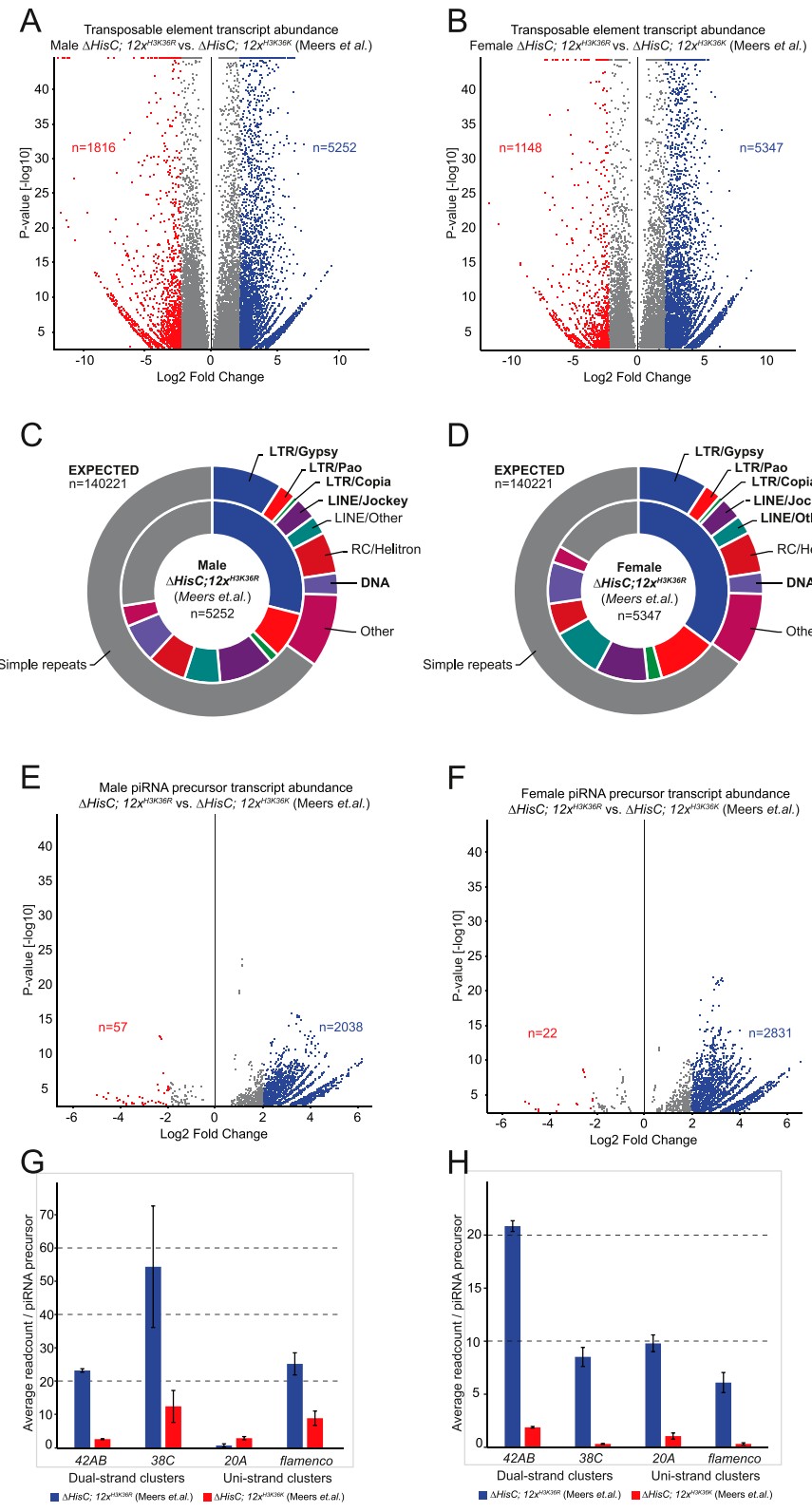

**Figure 4.   Increased transcription of transposons and piRNA clusters is observed in RNA sequencing of whole larvae.**
**(A)** Volcano plot showing differential transcription of transposable elements between $\Delta HisC$; $12x^{H3K36R}$ and $\Delta HisC$; $12x^{H3K36K}$ in males. Shown in blue are 5,252 transposons up-regulated by $\log_2$ fold change >2 ($P < 0.05$). **(B)** Volcano plot showing differential transcription of transposable elements between $\Delta HisC$; $12x^{H3K36R}$ and $\Delta HisC$; $12x^{H3K36K}$ in females. Shown in blue are 5,347 transposons up-regulated by $\log_2$ fold change >2 ($P < 0.05$). **(C)** Pie chart representing the expected ratio between transposon families based on genome abundance (outer circle) versus the observed transcription of 5,252 transposons in $\Delta HisC$; $12x^{H3K36R}$ males (inner circle). **(D)** Pie chart representing the expected ratio between transposon families based on genome abundance (outer circle) versus the observed transcription of 5,347 transposons in $\Delta HisC$; $12x^{H3K36R}$ females (inner circle). **(E)** Volcano plot showing differential transcription of piRNA precursor transcripts between $\Delta HisC$; $12x^{H3K36R}$ and $\Delta HisC$; $12x^{H3K36K}$ in males. Shown in blue are 2,038 piRNA precursors being up-regulated by $\log_2$ fold change >2 ($P < 0.05$). **(F)** Volcano plot showing differential transcription of piRNA precursor transcripts between $\Delta HisC$; $12x^{H3K36R}$ and $\Delta HisC$; $12x^{H3K36K}$ in females. Shown in blue are 2,831 piRNA precursors being up-regulated by $\log_2$ fold change >2 ($P < 0.05$). **(G)** Bar chart showing the male average number of reads mapping to piRNA precursors originating from dual-strand clusters *42AB* and *38C* (left) and uni-strand clusters *20A* and *flamenco* (right). The cluster average read number per piRNA precursor in $\Delta HisC$; $12x^{H3K36R}$ (blue), $\Delta HisC$; $12x^{H3K36K}$ (red), respectively: 23.17 and 2.51 for cluster *42AB*; 54.36 and 12.37 for cluster *38C*; 0.61 and 2.82 for cluster *20A*; 25.17 and 8.81 for the *flamenco* cluster. **(H)** Bar chart showing the female average number of reads mapping to piRNA precursors originating from dual-strand clusters *42AB* and *38C* (left) and uni-strand clusters *20A* and *flamenco* (right). The cluster average read number per piRNA precursor in $\Delta HisC$; $12x^{H3K36R}$ (blue), $\Delta HisC$; $12x^{H3K36K}$ (red), respectively: 20.84 and 1.88 for cluster *42AB*; 8.52 and 0.32 for cluster *38C*; 9.80 and 1.06 for cluster *20A*; 6.10 and 0.31 for the *flamenco* cluster. Error bars in (G, H) represent 95% confidence interval. The results in this figure were produced by analysing nuclear RNA sequencing data from whole larvae (Meers et al, 2017) using the same pipeline as used for our original data in brains.

abundance of RNA from piRNA clusters in both males (Fig 4E) and females (Fig 4F). When looking at RNAs from piRNA clusters, we observed strong transcription within dual-strand clusters in both males and females (Fig 4G and H). However, unlike in brains, we observed significantly elevated transcription of uni-strand piRNA clusters at comparable levels as dual-strand clusters. For example,

in males, we observe increased transcription of the *flamenco* cluster (Fig 4G) and, in females, both the *flamenco* and *20A* clusters (Fig 4H). This is presumably because of the presence of germ line cells where piRNA biosynthesis is constitutively active. Genes related to piRNA biosynthesis and dual-strand cluster activation are also being up-regulated. Significantly up-regulated genes ($P < 0.05$) in both sexes are: *AGO3*, *piwi*, *krimp*, *moon*, *rhi*, *fs(1)Yb*, *papi*, *egg*, *qin*, *SoYb*, *tud* and *vas* (Table S3). In addition, female larvae also exhibit significant up-regulation ($P < 0.05$) of: *del*, *BoYb*, *CG15930*, and *Kots* (Table S3). The only piRNA biosynthesis gene specifically up-regulated in males is *squash* (Table S3). To summarise, the impact of impaired H3K36 methylation on increased transposable element transcription and transcription of piRNA clusters can be detected in multiple tissues, sexes, and by independent experimental measurements.

# Discussion

We show here that compromised H3K36 methylation, likely through the loss of H3K36me2, causes increased transcription of transposable elements. The activation of transposable elements in turn triggers transcriptional activation of the PIWI machinery followed by transcriptional activation of dual-strand piRNA clusters. As the PIWI/piRNA system has a dampening effect on transposable-element transcription, we speculate that the observed effect seen in $\Delta HisC$; $12x^{H3K36R}$ is already lowered by the induced PIWI/piRNA system. The mechanism to increase the transcription of transposable elements remains elusive but does not depend on DNA methylation because *D. melanogaster* lacks 5-methylcytosine.

Repression and surveillance of selfish transposable elements in the germline is tremendously important to safeguard the integrity of the genome for future generations. Complex and adaptive regulatory systems, such as the ping-pong system first discovered in *D. melanogaster*, has evolved to efficiently target runaway transcription of transposons in germ cells. In recent years, however, the transcription of retrotransposons in somatic cells has shown to be tightly connected to cancer progression, neuronal defects, and brain development (Loreto & Pereira, 2017; Evans & Erwin, 2021). Studies in ageing suggest that loss of posttranslational histone marks in elder individuals results in increased transposon activity and susceptibility to cancer and neuronal degradation (Loreto & Pereira, 2017; Tsuji et al, 2021). Traditionally, posttranslational histone modifications with known links to transcriptional repression (e.g., H3K9 methylation) have been studied with regards to repression of transposon activity. However, a recent study suggested that the loss of H3K36 methylation, a histone modification previously linked to active transcription, caused transcriptional activation of transposable elements (Chaouch et al, 2021). The observation opens several central questions to be addressed. It was conducted by overexpression of the replication-independent histone variant H3.3K36M. This mutant protein has been shown to inhibit the enzymatic activity of K36-related histone methyltransferases (Lu et al, 2016). Whether the loss of the K36 methylation marks altogether and/or which function of any specific methyltransferase is responsible for observed phenotypes remained elusive. The latter is important because two out of the three K36-related methyltransferases may have non-histone

targets in addition to H3K36 which have severe effects on gene-regulatory functions (Dorafshan et al, 2019; Lindehell et al, 2021).

Our finding that transcriptional activation of transposons occurs in the replication-dependent histone mutant $\Delta HisC$; $12x^{H3K36R}$ but not in the histone variant H3.3 knock-out mutant $\Delta H3.3B$; $\Delta H3.3A$ shows that the replication-dependent histone H3 is the main isoform implicated in repression of transposable elements. Our result also reinforces the assertion that the lysine-36-to-methionine substitution is toxic to the involved methyltransferases (Lu et al, 2016) and would therefore disrupt methylation of non-substituted histones and potentially also other targets of the affected histone methyltransferases. From our analyses of *Set2*[1], *NSD*[ds46], and *ash1*[22]/*ash1*[9011] mutants individually, we find Set2 to be the main histone methyltransferase responsible for the repression of transposable elements with 58.9 percent of transposons being significantly up-regulated in $\Delta HisC$; $12x^{H3K36R}$ ($\log_2 FC > 2$, $P < 0.05$) also showing increased transposon activity in *Set2*[1] (Fig 2D and E). At first glance, this could have implicated trimethylated H3K36 as the main repressive mark because Set2 is the only known trimethyltransferase for H3K36 in *D. melanogaster*. However, ChIP-seq data show that H3K36me2 is significantly enriched in LINE and LTR transposable elements (Fig 2F). This result is supported by findings from polytene chromosome staining showing H3K36me2 enrichment in pericentromeric heterochromatin (Lindehell et al, 2021) which also are the genomic regions with the highest enrichment of LTR and LINE transposable elements (Fig 1C). Although correlative, this is in line with the notion that depletion of H3K36me2 in a mouse cell line largely recapitulates the transcriptional changes observed by H3.3K36M (Rajagopalan et al, 2021). Interestingly, recent evidence shows LTR transposable elements preferentially inserting into H3K36 methylation-rich regions (Cao et al, 2023), further corroborating the relationship between transposable elements and H3K36 methylation. Taken together, our results suggest that H3K36me2 is an important histone modification to control transposon activity. Our results further imply that when H3K36 methylation is impaired, transposons will start moving in the genome in somatic cells. Future studies are warranted to test this prediction.

Transcription of transposons in somatic cells is connected to cancer progression, neuronal defects and brain development (Baillie et al, 2011; Loreto & Pereira, 2017; Bedrosian et al, 2018; Evans & Erwin, 2021). At the same time, differences in DNA methylation of transposable elements were linked to their activity in the brain (Bedrosian et al, 2018). In mammalian cells, H3K36me2 has been shown to help recruiting DNA methyltransferase to chromatin and thus for the maintenance of DNA methylation (Weinberg et al, 2019; Chen et al, 2022). Disrupted H3K36 methylation was linked to various brain cancers and cognitive diseases (Kurotaki et al, 2002; Newbold & Mokbel, 2010; Xiao et al, 2021) and some of the observed consequences may thus be caused by its impact on transposon activity. Notably and important, DNA methylation is absent in *D. melanogaster*, whereas the repressive effect of H3K36 methylation on transposon–element activity remains. This argues that the cause–effect relationship between transposon activity, DNA methylation, and H3K36 methylation in mammalian cells remains to be carefully investigated.

The clustering of transcribed transposons shows patterns for *NSD*[ds46] and *ash1*[22]/*ash1*[9011] that are distinct but less pronounced

compared with $Set2^1$. $NSD^{ds46}$ and $ash1^{22}/ash1^{9011}$ mutants overlap with $\Delta HisC$; $12x^{H3K36R}$ by 14.6 and 4.3 percent, respectively. We find that division of labour between the histone methyltransferases in the repression of transposable elements is more dependent on what families of repeats are being repressed than on their genomic locations. Our results show that $Set2^1$ mutants transcribe a significantly increased amount of LTR repeats, the bulk of which belong to the Gypsy and Pao superfamilies. We observe an increase in LTR repeats in $ash1^{22}/ash1^{9011}$ and $NSD^{ds46}$ as well, albeit to a lower degree. This underscores the importance of Set2-driven methylation of H3K36 in keeping this group of repeats transcriptionally inactive. LINE family Jockey repeats are also transcribed in all three conditions, but most significantly in $NSD^{ds46}$ where 19.0% of all transcribed transposons belong to the Jockey family repeats (Table S1). The transposon-activity profile of $ash1^{22}/ash1^{9011}$ is the least biased but interestingly has the biggest relative increase in LTR/Copia repeats with 6.25% compared with $Set2^1$ (5.03%) and $NSD^{ds46}$ (1.67%) (Table S1). We note that because of multi-mapping of reads, we cannot determine the exact origin of the transcribed transposable elements. However, we show that the bulk of transposable elements originate from intergenic regions and the observed up-regulation of transposable elements is not a result of transcriptional activation or increased transcription of transposon harbouring genes (Fig 1D).

The largely overlapping increase in transcription of repeated elements after knock-out of H3K36-specific methyltransferases (particularly Set2) and H3K36R replacement (replication-dependent H3.2K36R) argues that it is the methylation of H3K36 that keeps transposable elements transcriptionally inactive in somatic cells. However, the histone replacement approach has limitations. First, H3K36R replacement effectively abolishes all lysine modifications including acetylation. It also may affect post-translational modifications of neighbouring amino acids. Thus, H3K36R is less effectively methylated at lysine 27 (H3K27) by the Polycomb Repressive Complex 2 (PRC2) in vitro (Jani et al, 2019; Finogenova et al, 2020). Likewise, flies with mutations that replace both the replication-independent H3.3K36R and the replication-dependent H3K36R display stochastic transcription of homeotic genes in cells where they are supposed to be repressed by the Polycomb system (Salzler et al, 2023). It is thought that H3K36R replacement inhibits PRC2-mediated methylation and may be structurally analogous to inhibition of PRC2 by di or trimethylated H3K36 (Jani et al, 2019; Finogenova et al, 2020). It seems unlikely that the increased transcription of transposable elements in H3K36R larvae is caused by impaired PRC2 activity. This would require that the loss of H3K36 methyltransferases (the loss of PRC2 inhibitors) and the H3K36R replacement (PRC2 inhibition) lead to the same effect. Whether transcriptional activation of transposable elements in somatic cells of H3K36R mutants may cause erroneous activation of homeotic genes remains an interesting open question.

To combat runaway transcription of transposable elements, the PIWI/piRNA system has evolved utilizing Argonaut endoribonucleases binding piRNAs to specifically target transcribed transposons (Czech & Hannon, 2016). This mechanism is mainly studied in the germline, but piRNA expression in somatic cells exists and has been shown in multiple species, including *D. melanogaster*

(Loreto & Pereira, 2017). Our study argues that all necessary components for piRNA surveillance are expressed in the *D. melanogaster* brain if the H3K36 methylation machinery is compromised and transposable elements are activated. We show that in $\Delta HisC$; $12x^{H3K36R}$ and in $Set2^1$ and $NSD^{ds46}$ knock-outs there is significant transcription of the piRNA clusters, whereas the transcription is less pronounced in $ash1^{22}/ash1^{9011}$. Functional piRNA production is best examined using small RNA sequencing, which was used by Chaouch et al (2021) to confirm the expression of piRNAs upon overexpression of H3.3K36M. We have analysed RNA-seq with a paired-end read length of 150 bp. Still, considering the same loci became activated upon overexpression of H3.3K36M and, in $\Delta HisC$; $12x^{H3K36R}$, and in $Set2^1$ and $NSD^{ds46}$, it seems likely that in all cases, piRNAs are formed. The PIWI/piRNA system is not fully understood but one prevailing model is that piRNA clusters act as traps for transposons that once inserted silence transposable elements in *trans* (Zanni et al, 2013). This model supports the idea that piRNA transcriptional activation would be dependent on transposable element expression levels. We hypothesize this to be the reason for the increased levels of piRNA precursors in $\Delta HisC$; $12x^{H3K36R}$ and $Set2^1$ where transposon transcript levels are highest, that is, the PIWI machinery is induced by the increased transcript levels of transposable elements and this induction is followed by piRNA production. We also conclude that dual-strand clusters, that in the germline relies on H3K9me3, Rhino, Moonshiner, and Deadlock for their transcription, are more sensitive to increased level of transposon RNA. The imbalance between cluster-type activation might explain why transposon activity is present in the somatic transcriptome unlike in the germline where the PIWI/piRNA system is hyperactivated and the transposon silencing is complete. For PIWI/piRNA biosynthesis-related genes, we observe increased transcription in $\Delta HisC$; $12x^{H3K36R}$ and $Set2^1$. Importantly, we only observe significant transcriptional activation of the Argonaut family genes under these conditions, *Aubergine* and *Argonaut-3* for $Set2^1$ and $\Delta HisC$; $12x^{H3K36R}$ and also *piwi* in $\Delta HisC$; $12x^{H3K36R}$. We note transcriptional activation of Tudor domain containing protein-encoding genes *Kots* and *CG15930*. *Kots* is up-regulated to the same degree as *Krimper* which has previously been identified as a major factor for transposon repression in H3K36M histone-replacement flies (Chaouch et al, 2021). *Kots* encodes a germline-specific nuage protein that is required to ensure proper piRNA production from both dual-strand and uni-strand piRNA clusters (Lim & Kai, 2007; Lim et al, 2022). Assuming the somatic expression of *Kots* retains a similar function as in the germline, we speculate that *Kots* is a key contributor for the observed low-level activation of uni-strand clusters in $\Delta HisC$; $12x^{H3K36R}$, whereas at the same time enhancing the RNA production from Rhino-dependent dual-strand piRNA clusters. *CG15930* is the single most up-regulated Tudor domain containing transcript in $\Delta HisC$; $12x^{H3K36R}$ and has been shown to promote male sexual identity in the germline (Primus et al, 2019). It is plausible that *CG15930* is connected to the PIWI/piRNA pathway and antagonizes transposon activity; however, this remains to be tested.

To summarize, here we show that methylation of lysine 36 on replication-dependent histone H3 plays a central role in keeping transposable elements transcriptionally inactive in somatic cells. We also demonstrate that different methyltransferases involved in

the process have a distinct contribution in suppressing different transposon families and that Set2-driven dimethylation of H3K36 appears to be the most potent suppressor. Because *D. melanogaster* lacks DNA methylation, the molecular mechanism does not rely on this process, which in turn suggests that DNA methylation of repeated elements in mammals may be a consequence of their transcriptional inactivity rather than the primary cause. When somatic H3K36 methylation is lost or impaired and transposable elements become transcribed, piRNA surveillance is activated and the degree of activation mirrors the abundance of transposable elements expressed. For reasons that require further studies, this leads to preferential activation of dual-strand Rhino-dependent piRNA clusters.

## Materials and Methods

### Sequencing data

The RNA sequencing data from dissected *D. melanogaster* third instar larvae male brains are described in a previous study Lindehell et al (2021) and the data are available in the Gene Expression Omnibus under the extension GSE166934. The data are based on five to seven biological replicates each of Oregon R, $ash1^{22}/ash1^{9011}$, $NSD^{ds46}$, $Set2^1$, $\Delta HisC; 12x^{H3K36R}$, $\Delta HisC; 12x^{H3K36K}$, and $\Delta H3.3B; \Delta H3.3A$. The mutant alleles have been described previously (Larschan et al, 2007; Hödl & Basler, 2009; McKay et al, 2015; Dorafshan et al, 2019; Lindehell et al, 2021).

The RNA sequencing data from eye discs after overexpression of H3.3K36M was produced by Chaouch et al (2021) and is accessible under the extension GSE140979. Nuclear RNA sequencing data from whole third instar larvae were produced by Meers et al (2017) and is available under the extension GSE96922. For ChIP-seq of H3K36 me1, me2, and me3, modEncode data from mixed sex adult heads were used and are accessible under the extensions: GSE47279, GSE47336, and GSE47280 (Celniker et al, 2009).

### Bioinformatics

The RNA sequencing data were processed as described in Lindehell et al (2021) and BAM files were imported into SeqMonk v.1.47 (RRID: SCR_001913) using *D. melanogaster* genome release BDGP6. For transposon analysis, uniquely aligned reads (mapQ > 0) were quantified within Repeatmasker (v.4.0.6) annotations (Smit et al, 2015). For analysis of RNAs originating from piRNA clusters, piRNA annotations were taken from piRNAdb (Piuco & Galante, 2021) alongside piRNA cluster coordinates from Brennecke et al (2007). Differential expression analysis of genes was performed using DeSeq2 with default parameters and normal log fold change shrinkage (Love et al, 2014). Stringent filtering of the differential expression analysis was performed not to include transposons/precursor piRNAs with significantly changed transcription between two independently sequenced Oregon R runs (Five to six replicates each) and between simultaneously sequenced $\Delta HisC; 12x^{H3K36K}$ and Oregon R (four and five replicates each). For $\Delta HisC; 12x^{H3K36R}$, the number of transposons filtered out was 1,694. For $Set2^1$, $NSD^{ds46}$,

$ash1^{22}/ash1^{9011}$, and $\Delta H3.3B; \Delta H3.3A$ the number of filtered out transposons were 2,354, 3,641, 2,586, and 3,799, respectively. The whole larvae nuclear RNA-seq results are presented unfiltered because of insufficient number of biological and technical replicates. ChIP-seq data were assigned to repeats using Repeatmasker (v.4.0.6) annotations (Smit et al, 2015) in SeqMonk v1.47 (RRID: SCR_001913) and genome build dm3. Average enrichment for all H3K36 methylation states and associated input samples (two replicates each) was calculated to generate sample/input ratios for exons, simple repeats, LINE and LTR-repeats, and 10,000 randomly selected regions of 80 base pairs each. Enrichments of piRNA biosynthesis-related gene ontology terms were calculated using DAVID (Huang da et al, 2009; Sherman et al, 2022) on the top 500 most up-regulated genes from the differential transcription analysis between $\Delta HisC; 12x^{H3K36R}$ and $\Delta HisC; 12x^{H3K36K}$.

## Supplementary Information

## Acknowledgements

We thank the Science for Life Laboratory, Stockholm, Sweden; the National Genomics Infrastructure (NGI), Stockholm, Sweden; and UPPMAX, Uppsala, Sweden, for providing the computational infrastructure. This work was supported by grants from the Swedish Research Council (grant numbers 2020-03561 to J Larsson and 2021-04435 to YB Schwartz) and the Swedish Cancer Foundation (grant number 20 0779 PjF to J Larsson and 22 2285 Pj to YB Schwartz). Funding for open access charge: the Swedish Research Council (grant numbers 2020-03561).

### Author Contributions

H Lindehell: conceptualization, data curation, software, formal analysis, validation, investigation, visualization, methodology, and writing—original draft, review, and editing.
YB Schwartz: conceptualization, formal analysis, funding acquisition, validation, methodology, and writing—review and editing.
J Larsson: conceptualization, formal analysis, supervision, funding acquisition, validation, methodology, project administration, and writing—review and editing.

### Conflict of Interest Statement

The authors declare that they have no conflict of interest.

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
