## [Reviewer comments · Life Science Alliance]

Life Science Alliance

Methylation of lysine 36 on histone H3 is required to control transposon activities in somatic cells

Henrik Lindehell, Yuri Schwartz, and Jan Larsson

DOI: <https://doi.org/10.26508/lsa.202201832>

Corresponding author(s): Jan Larsson, Umeå University

Review Timeline:

Submission Date:	2022-11-16
Editorial Decision:	2022-12-15
Revision Received:	2023-04-03
Editorial Decision:	2023-05-01
Revision Received:	2023-05-03
Accepted:	2023-05-03

Transaction Report:

December 15, 2022

Re: Life Science Alliance manuscript #LSA-2022-01832-T

Prof. Jan Larsson
Umeå University
Molecular Biology
Dept of Molecular Biology
Umeå SE90187
Sweden

Dear Dr. Larsson,

Thank you for submitting your manuscript entitled "Methylation of lysine 36 on histone H3 is required to control transposon activities in somatic cells" to Life Science Alliance. The manuscript was assessed by expert reviewers, whose comments are appended to this letter. We invite you to submit a revised manuscript addressing the Reviewer comments.

Thank you for this interesting contribution to Life Science Alliance. We are looking forward to receiving your revised manuscript.

Sincerely,

B. MANUSCRIPT ORGANIZATION AND FORMATTING:

Reviewer #1 (Comments to the Authors (Required)):

In the manuscript entitled "Methylation of lysine 36 on histone H3 is required to control transposon activities in somatic cells" the authors show that upon replacement of the replication-dependent histone H3 by an H3K36R mutant, transposable elements are derepressed in somatic cells of the brain. They find that null mutants of the three histone methyltransferases (HMT's) known to methylate H3K36 (Ash1, Set2, NSD) recapitulate only partially the effect of the K36R mutation, with Set2 loss showing the strongest overlap. The deregulation of transposons is correlated with an increased expression of piRNAs (preferentially from the dual-strand piRNA cluster) and genes implicated in transposons silencing in the germline (piRNA pathway), suggesting that this pathway can also be activated at least in part in non-germline somatic cells.

The comparison of H3K36R histone replacement to null mutants of the 3 HMTs known to methylate K36 allows to clarify the contribution of K36 methylation and other potential functions of the HMTs. Here the authors show that the loss of the three HMTs recapitulates the effect on TE expression detected in the H3K36R background.

General comments

- To our knowledge, to date derepression of transposable elements (TE's) in non-germ cells in *Drosophila* has been described only once, upon expression of histone H3.3K27M and H3.3K36M in eye imaginal disks. In this system, TE's de-repression is linked to activation of the piRNA pathway (Chaouch, 2021).
- Limitation of the study. The authors use their own published RNA-seq data sets (Lindehell, 2021) and the distribution of H3K36me1/2/3 in control conditions (ModEncode) for the analysis. They do not know to which extent the H3K36me1/2/3 amount, the distribution of those methyl-marks genome-wide and in particular on the various copies of each transposon type are affected upon depletion of the three HMTs.
- The authors assume that derepressed copies of transposons are mostly located in constitutive heterochromatin (pericentromeric, Y chromosome) where the majority of H3K36me2 and NSD are located, as they have previously shown (Lindehell, 2021). The H3K36me2 enrichment at the most variable TEs (LTR/LINE) shown in Fig. 2F is also compatible with preferential distribution to constitutive heterochromatin. However, they propose that transposon de-repression is mostly dependent on H3K36me2 catalyzed by Set2, which is rather expected to be found at RNA-polymerase II (polII) transcribed genes, since Set2's localization and activity depends on phosphorylation of PolII's CTD. The authors should discuss potential direct and indirect effects of Set2 loss on H3K36 methylation. Could the authors compare the extent of de-repression for TEs, to the fraction of TE copies located within active and inactive genes?
- Since piRNA gene clusters are also enriched in heterochromatin regions, could the effect on piRNA gene expression directly depend on the loss of H3K36 methylation genome wide, rather than be a consequence of TE de-repression? This could be particularly true for the HMT mutants that show a mild phenotype in terms of expression of piwi/piRNA biogenesis genes, whereas in the H3K36R mutant where the extent of H3K36me loss is expected to be much more prominent, various genes in the pathway are clearly induced.

Major points

- Abstract: The authors did not show that H3K36 methylation (directly) represses transcription of transposable elements, but that the loss of H3K36me induces de-repression of TE. They also did not show that the effect is mostly dependent on the loss of H3K36me2 at transposable elements, but only that de-repressed transposons tend to have mostly H3K36me2 in control conditions, which is expected since most copies are within constitutive heterochromatin.
- Introduction: since H3K36 appears to have different predominant functions in different organisms, the authors should distinguish what is known from other organisms (mammals/plants) and what is known in *Drosophila*. For example, TE de-repression and piRNA pathway activation outside the germline.
- Fig. 1. Localization of de-repressed Transposons: the authors should clarify that they cannot identify which of the copies of each transposon present in the genome are de-repressed. Most TE copies are within pericentric heterochromatin or on the Y chromosome, but in principle the authors cannot exclude that the upregulation is contributed from the copies present on chromosome arms. Are the RC/helitron copies, which are not derepressed, not enriched at constitutive heterochromatin at all? Where are they located relative to genes?
- Fig.2 Readers should be alerted that the Δ HisC; 12xH3K36K strain differs in transcription profile from OregonR. OregonR is the control for Set2, Ash1, NSD, H3.3B/A deletion strains, but K36K is the control for K36R. Comparison of the two experiments is complicated by the fact that the 2 controls are very different. The authors mentioned that for transposon analysis they excluded all transposons that have significantly changed expression between K36K and OregonR. They should mention how many

transposons were dropped and how many were used for the comparison. Fig. 2E "Expected" should be corrected accordingly. Table EV1 should be updated with an additional column with only the TE's used for comparison. In Fig. 2F, authors should rather compare de-repressed transposons, to unaffected ones and to down-regulated ones to address the significance of the H3K36 methylation state.

- Fig. 3. Are piRNA genes over-represented among upregulated genes in the different genetic backgrounds? (GO/GSEA analysis) .ie. is piRNA pathway specifically activated or is the observed increase a consequence of the general genome-wide expression increase at protein-coding genes?
- The authors should reanalyze H3K36R mutant larval RNAseq dataset from Meers (2017) Elife (This is the only other H3K36R fly dataset, they also show that many low expressing genes get upregulated in K36R mutants) to see if they observe a similar effect - Both similarities and differences will prove to be interesting to the readers.
- Discussion: authors should stress the limitations of the study. For example, autonomously generated transposon transcripts would be better quantified from 5'cap RNA-seq and not from bulk RNA-seq in which the reads can be contributed by introns (often containing TE's), also functional piRNA production can only be shown by small RNA sequencing. Possible indirect effects should also be discussed.

Minor points

- All figure legends incorrectly spell "Volcano" plots as "Vulcano" plot .
- Color code in all main and EV figures should be consistent (it even changes between panel of the same figure!) and clear (Fig. 1C 2 different blues cannot be distinguished)
- Fig 1. (A) Plot title. Replace "vs WildType" to "vs Δ HisC; 12xH3K36K" and use color code presented in 1B for TE's selected for the analysis. $P < 0.05$ should be at 1.3 on the Y-scale (B) Expected TE distribution does not correspond to Table EV1. Example: other. Add $N=4088$ and $N=140221$ to both categories. Same is true for Fig. EV1(C) Should represent TE's upregulated according to legend: LTR/Gypsy, Pao and Copia, LINE/Jockey and other as well as DNA but not RC or other. Why is the heterochromatic part of the X chromosome that small in this representation? (D) Is "wild type" used as reference here, or Oregon R? Same question for Fig. 2A/B/C.
- Fig. 2. (A/B/C) WT is Oregon R. Add color code from 1B. Y scale. (D) $N=?$ How many of the 4088 TE's are analyzed here? If these are only upregulated TEs then why does the scale representing \log_2 FC ranging from -1.5 to +1.5? (E) color code consistency.
- Fig. 3. (A/B) Number in Figure are total mis-regulated piRNA genes, rather use upregulated ones.
- Table 1: Mark all Tudor domain proteins, CG15930 and kots also have tudor domains
- Fig EV2. Color code consistency again. Represent the same relevant TE's as in Fig1C. Is there a difference for LINE and/or Simple repeats on the X in the K36R mutant, or is this due to an unfortunate choice of colors?

Reviewer #2 (Comments to the Authors (Required)):

Summary:

Lindeh et al in this manuscript discuss the role of methylation on lysine 36 on histone H3 and its requirement to control transposon activities in somatic cells. To do so authors have assessed transcript levels of transposable elements in brains dissected from third instar male larvae of Lysine to Arginine replacement Δ HisC; 12xH3K36R mutant and the reciprocal control. In agreement with Chaouch et al., 2021, authors have identified transcriptional activation of TE. Majority of which being Long Terminal Repeats (LTR) of Gypsy, Pao and Copia and Jockey family line repeats as previously seen. The genomic location of these TE are also in line with what was previously shown as the bulk being present in the pericentromeric region of all chromosomes. Authors take some thorough and interesting approach in profiling the status of TE activation in 3 of the methyltransferase mutants involved in methylation of H3 K36 and make a comparison with the K36R mutants. They find division of labor whereby each methyltransferase contributes to the process of TE activation the highest being Set2 and the other 2 less with more complementary functions. In an attempt to explain some mechanism of TE regulation authors assess TE production in different mutants and find a correlation between the increase of TE production in different methyltransferase mutants and K36R mutant. Furthermore, by analyzing chip-seq data on k36me1, me2, me3 they assess the enrichment of these marks in TE and find that highest K36me2 is found on LINE and LTR regions, suggesting me2 mark of K36 loaded by the 3 methyltransferases to be involved in the prevention of transposable element transcription. Finally, authors assess whether piRNA expression is also upregulated in Δ HisC; 12xH3K36R and the three methyltransferase mutants. They find that loss of h3 K36 methylation triggers piRNA production from dual strand piRNA clusters and further observe transcriptional activation of almost all genes linked to piRNA biogenesis.

The manuscript is very well written and it further reinforces that H3k36 is necessary for repression of TEs in somatic cells and that mutations within this lysine may cause CNS diseases and cancer as shown in literature. However, the proposed mechanism that it is through K36me2 loss is highly possible but not clearly shown/described in the manuscript and should be further clarified/explained. Special clarification should be granted when it comes to comparisons with H3K36M seq data that are done on eye discs whereas majority of transcriptomic data are performed on 3rd instar larval brains.

Here are some points that require further explanation:

- 1- The authors show convincing data on transcriptional activation of transposable elements in absence of methylation on the replication coupled histone H3, presuming that Δ HisC; 12xH3K36R replacement loses k36 methylation mark. It will be easier to follow for the reader if this is clearly explained in the text even if it is previously published and the references are included in the manuscript. In other words, it will be good to have an overview of the me1,2,3 methylation state of this mutant that they have

used right at the beginning of the manuscript.

2- Did the authors test K36R me3, me2, me1 level in those methyltransferase mutants (set2, NSD and ash)? If not, has it been shown previously in the literature? can authors explain what is known here or what has been done?

3- In page 6 of the manuscript (line 162) authors ask whether the TE activation in K36R is linked to one of the methylation states of k36. They mention that they have done that by analyzing seq data for mono-, di- and trimethylated H3K36 from adult fly brains. Why is so? When all the transcriptomic data is done on third instar larval brains.

4- Related to the same question above in material and methods referring to data produced in figure 2 (E, F, EV1), authors mention page 13 (line 384) that the RNA seq is from adult heads (GSE140979) (Chaouch et al., 2021) while all chip-seq data performed in this study is done on eye discs. Can the authors explain? This also raises the question about tissue specificity of these marks as these methylation marks are measured in eye discs, therefore making the argument of the TE activation seen in larval brains of K36R due to loss of me2 not completely accurate. Additionally, when such comparison is done in different tissues the CNS context of this observation should be reconsidered as this is tested in 2 different tissues and should be clearly stated and elaborated in text. Also, can authors explain why they chose to look into the methylation mark levels if it varies amongst different transposons in H3k36M and not the wild type H3.3 overexpression as well to have the full picture.

5- In addition I think clear distinction should be also made that sequence data of H3k36M used in the study is from overexpression studies and it might not recapitulate completely the situation with Δ HisC; 12xH3K36R replacement used in this current study although it is interesting that they share somehow similar features (Fig EV1). It will be good to clearly mention this in the text.

Few minor comments:

1- I think it should be clearly stated why male larval brains were used in the study?

2- Page 11 Discussion (line 326, it seems unlikely to line 329 same effect) is unclear and requires some sentence breakdown and more explanation.

3- I recommend moving Fig EV3 from supplement to main as it will be nice to see the whole picture of piRNA expression in all three methyl transferases.

Reviewer #3 (Comments to the Authors (Required)):

The manuscript 'Methylation of lysine 36 on histone H3 is required to control transposon activities in somatic cells' dissects the effect of three histone H3 K36 methyltransferases, Set2, NSD and Ash1, in the eye of *Drosophila melanogaster*. As previously described for loss of H3K36 methylation in *Drosophila* eyes, the authors observe an upregulation of transposon transcripts (Figure 1). Figure 2 shows that the upregulation of TE transcripts is strongest in knock-down of Set2, the only MTase that tri-methylates H3K36. Knockdown of NSD and Ash1, which deposit mono- and di-methylation on H3K36, show minor effects on TE transcript abundance. The authors observe increased transcript abundance for mostly LTR-derived sequences in Set2 knock-down (Fig. 2E). However, such LTR elements do not show enrichment for H3K36me3, which is deposited by Set2 (Fig. 2F). Both LTR and Non-LTR (LINE) elements show enrichment of H3K36me2 by ChIP. In Fig. 2F enrichment of H3K36me2 and me3 is mostly observed across exons.

The authors then start talking about PIWI-interacting RNAs (piRNAs). Experimentally they seem to have compared short RNA-seq reads, which technically represent parts of longer RNAs (>200nt for standard RNAseq protocols), to sequences in a piRNA-database. The authors wrongly call their analysis 'piRNA analysis'. Figure 3 and the corresponding parts of the manuscript and method exhibit a complete lack of understanding of what piRNAs are: PIWI interacting RNAs (piRNAs) are small RNAs that are defined by their physical association with PIWI proteins (Girard et al., 2006). To interrogate piRNAs, the authors must (1) show the presence of one of the three fly PIWI proteins, which are germline-restricted genes under physiological conditions, (2) immunoprecipitate the expressed PIWI protein, and (3) prepare the associated small RNAs for next-generation sequencing. I recommend the authors to familiarize themselves with piRNA biology: Reviews and primary literature from the Hannon, Zamore, Brennecke, Aravin and Siomi and other labs.

This manuscript seems to be based on a poor understanding of transposons and a complete misunderstanding of piRNA biology. The tested hypotheses, experimental approaches and interpretations are fundamentally flawed.

Major points:

Fig. 1 The main message has already been described in Chaouch et al., 2021.

Minor issues:

The authors observe changes in steady state RNA levels not activation of transcription. The title of this figure should be rephrased accordingly.

Fig. 1C: This panel indicates the genomic location of all possible TE insertions that could be responsible for the observed increase in TE transcript levels. Due to multimappability of TE-derived reads in RNA-seq, the authors cannot for sure determine their genomic origin. The interpretation of this depiction should be rephrased accordingly or remove this figure panel.

Fig. 2. is the first figure that contains novelty and could be merged with Fig 1.

Fig. 2F shows enrichment of H3K36me2 and m3 mostly on 'exons'. Are these protein coding exons or all exons? A differential expression of these regions should be shown, and they might be the most prominent changes of the transcriptome.

Figure 3. The presented 'piRNA' data are an artifact based on a complete misunderstanding of the fundamental definition of piRNAs. I recommend the authors to start familiarizing themselves with the basic literature on piRNA biology.

Comments and measures on behalf of the reviewers' comments

Reviewer #1

In the manuscript entitled "Methylation of lysine 36 on histone H3 is required to control transposon activities in somatic cells" the authors show that upon replacement of the replication-dependent histone H3 by an H3K36R mutant, transposable elements are derepressed in somatic cells of the brain. They find that null mutants of the three histone methyltransferases (HMT's) known to methylate H3K36 (Ash1, Set2, NSD) recapitulate only partially the effect of the K36R mutation, with Set2 loss showing the strongest overlap. The deregulation of transposons is correlated with an increased expression of piRNAs (preferentially from the dual-strand piRNA cluster) and genes implicated in transposons silencing in the germline (piRNA pathway), suggesting that this pathway can also be activated at least in part in non-germline somatic cells.

The comparison of H3K36R histone replacement to null mutants of the 3 HMTs known to methylate K36 allows to clarify the contribution of K36 methylation and other potential functions of the HMTs. Here the authors show that the loss of the three HMTs recapitulates the effect on TE expression detected in the H3K36R background.

Reviewer 1 - GENERAL COMMENTS

1.1 To our knowledge, to date derepression of transposable elements (TE's) in non-germ cells in Drosophila has been described only once, upon expression of histone H3.3K27M and H3.3K36M in eye imaginal disks. In this system, TE's de-repression is linked to activation of the piRNA pathway (Chaouch, 2021).

We agree with the comment.

1.2 Limitation of the study. The authors use their own published RNA-seq data sets (Lindehell, 2021) and the distribution of H3K36me1/2/3 in control conditions (ModEncode) for the analysis. They do not know to which extent the H3K36me1/2/3 amount, the distribution of those methyl-marks genome-wide and in particular on the various copies of each transposon type are affected upon depletion of the three HMTs.”

We value this comment and since we have not performed ChIP-sequencing in the specific mutants it is true we lack high resolution mapping data of H3K36 methylation. However, we have previously performed polytene chromosome staining showing H3K36me2 enrichment in pericentromeric heterochromatin (Lindehell et al., 2021). We have also published western blot analysis for the different methylation states in methyltransferase mutant conditions which provides information on depletion for each of these marks (Dorafshan et al., 2019; Lindehell et al., 2021). In this revised version, we believe that that the limitations are acknowledged and the conclusions drawn in the manuscript are valid. We have clarified this to the readers and have included references to earlier work from our groups where appropriate.

1.3 The authors assume that derepressed copies of transposons are mostly located in constitutive heterochromatin (pericentromeric, Y chromosome) where the majority of H3K36me2 and NSD are located, as they have previously shown (Lindehell, 2021). The H3K36me2 enrichment at the most variable TEs (LTR/LINE) shown in Fig. 2F is also compatible with preferential distribution to constitutive heterochromatin. However, they propose that transposon de-repression is mostly dependent on H3K36me2 catalyzed by Set2, which is rather expected to be found at RNA-polymerase II (polII) transcribed genes, since Set2's localization and activity depends on phosphorylation of PolII's CTD. The authors should discuss potential direct and indirect effects of Set2 loss on H3K36 methylation. Could the authors compare the extent of de-repression for TEs, to the fraction of TE copies located within active and inactive genes?.

We appreciate this comment/suggestion and have tested this by mapping transposon reads within active, inactive genes and intergenic regions. The data is presented in the new panel Fig 1D. The following text describes the results, line 135-144:

“To verify that the increase in transposable elements is not the result of increased transcription of genes harbouring transposable elements we quantified LTR and LINE elements within active and inactive genes as well as within intergenic regions. We found that the majority (71.7%) of upregulated transposons in $\Delta HisC$; $12x^{H3K36R}$ is associated with intergenic regions and, only 7.3% is associated with genes transcribed in the wild type animals (Fig 1D). Genes that are silent in the wild type animals accommodate 21% of upregulated LTR and LINE elements in $\Delta HisC$; $12x^{H3K36R}$ and within this group only 10.5% of the transposons localize to genes that are transcriptionally activated in $\Delta HisC$; $12x^{H3K36R}$ (Fig 1D). Therefore, we conclude that an increased transcription of genes' harbouring transposons is not responsible for the dramatic increase in transposon transcription in $\Delta HisC$; $12x^{H3K36R}$.”

1.4 Since piRNA gene clusters are also enriched in heterochromatin regions, could the effect on piRNA gene expression directly depend on the loss of H3K36 methylation genome wide, rather than be a consequence of TE de-repression? This could be particularly true

for the HMT mutants that show a mild phenotype in terms of expression of piwi/piRNA biogenesis genes, whereas in the H3K36R mutant where the extent of H3K36me loss is expected to be much more prominent, various genes in the pathway are clearly induced.

We have considered this, and our current data does not resolve the order of events. However, since dual-strand piRNA clusters require PIWI machinery components such as Rhino, Moonshiner and Deadlock for their transcription, we speculate that the most parsimonious explanation is that the piRNA defences are being mounted to counteract transposon activity. This is discussed in the discussion section lines 291-295 and 395-412.

Reviewer 1 – Major points

1.5 Abstract: The authors did not show that H3K36 methylation (directly) represses transcription of transposable elements, but that the loss of H3K36me induces de-repression of TE. They also did not show that the effect is mostly dependent on the loss of H3K36me2 at transposable elements, but only that de-repressed transposons tend to have mostly H3K36me2 in control conditions, which is expected since most copies are within constitutive heterochromatin.

We agree with the statement. We have checked our formulations and we believe our abstract doesn't include any unwarranted claims.

1.6 Introduction: since H3K36 appears to have different predominant functions in different organisms, the authors should distinguish what is known from other organisms (mammals/plants) and what is known in drosophila. For example, TE de-repression and piRNA pathway activation outside the germline.

This is a good comment and we have checked and made changes to the introduction to make it clearer what organism are used in the various studies referred to.

1.7 Fig. 1. Localization of de-repressed Transposons: the authors should clarify that they cannot identify which of the copies of each transposon present in the genome are de-repressed. Most TE copies are within pericentric heterochromatin or on the Y chromosome, but in principle the authors cannot exclude that the upregulation is contributed from the copies present on chromosome arms. Are the RC/helitron copies, which are not derepressed, not enriched at constitutive heterochromatin at all? Where are they located relative to genes?

We appreciate this question which closely relates to point 1.3. As mentioned above, in this revised version we have mapped transposon reads within active, inactive genes and intergenic regions and the data is presented in the new panel Fig 1D. It is true that for most transposons, because of varying copy number, we cannot give exact origins. However, with most of the LINE and LTR elements mapping to intergenic regions within pericentromeric and telomeric heterochromatin and not to genes, we believe these regions to be the premier origin of transposon activity. As for RC/helitrons, these are also preferentially enriched within pericentromeric heterochromatin and in telomeres and the intragenic

helitrons tend to be enriched towards the 3-prime end of genes. We have added the following statement to the discussion to clarify the multimapping problem dealing with repeats, line 366-370:

“We note that due to multimapping of reads we cannot determine the exact origin of transcribed transposable elements. However, we show that the bulk of transposable elements originate from intergenic regions and the observed upregulation of transposable elements is not a result of transcriptional activation or increased transcription of transposon harbouring genes (Fig 1D).”

1.8 Fig.2 Readers should be alerted that the Δ HisC; 12xH3K36K strain differs in transcription profile from OregonR. OregonR is the control for Set2, Ash1, NSD, H3.3B/A deletion strains, but K36K is the control for K36R. Comparison of the two experiments is complicated by the fact that the 2 controls are very different. The authors mentioned that for transposon analysis they excluded all transposons that have significantly changed expression between K36K and OregonR. They should mention how many transposons were dropped and how many were used for the comparison. Fig. 2E "Expected" should be corrected accordingly. Table EV1 should be updated with an additional column with only the TE's used for comparison. In Fig. 2F, authors should rather compare de-repressed transposons, to unaffected ones and to down-regulated ones to address the significance of the H3K36 methylation state.

We appreciate this comment and have clarified the usage of the histone wild type H3K36K control where appropriate. We have also included the number of transposons being filtered out to the Materials and Methods section (line 464-467). Table S1 does in fact show only transposons that passed the filter and are used for analysis. This comment also prompted us to produce figure 1D to show the proportional localization of upregulated LTR and LINE elements within intergenic regions, wild type transcribed genes and wild type untranscribed genes. Please note that the additional datasets added in this revised version (from Meers et al 2017) are unfiltered due to insufficient number of replicates. Still this analysis corroborates our findings and strengthens the conclusion.

1.9 Fig. 3. Are piRNA genes over-represented among upregulated genes in the different genetic backgrounds? (GO/GSEA analysis) .ie. is piRNA pathway specifically activated or is the observed increase a consequence of the general genome-wide expression increase at protein-coding genes?

We appreciate this comment and have included this analysis, Table S2. Line 252-253:

“Geno ontology for the top 500 most upregulated genes ($p < 0.05$) in Δ HisC; 12xH3K36R showed significant enrichment for genes related to piRNA metabolic processes (Table S2).”

1.10 The authors should reanalyze H3K36R mutant larval RNAseq dataset from Meers (2017) Elife (This is the only other H3K36R fly dataset, they also show that many low expressing genes get upregulated in K36R mutants) to see if they observe a similar effect - Both similarities and differences will prove to be interesting to the readers.

We are thankful for this suggestion and have analysed the data from Meers *et al* (2017), separating males and females. The results are shown in Figure 4 and with the accompanying text (lines 266-288) and figure legend. The results support our findings. We observe increased transposon transcription in both sexes (Fig 4A-B) with similar relative abundances between different classes of transposons as observed in the brain and in the eye discs (Fig 4C-D, compared to Fig 1B, Fig S1). We also observe increased abundance of RNA from piRNA clusters in both males (Fig 4E) and females (Fig 4F). When looking at RNAs from piRNA clusters, we observe strong transcription within dual-strand clusters in both males and females (Fig 4G-H). However, unlike in brains we observe significantly elevated transcription of uni-strand piRNA clusters at comparable levels as dual strand clusters. For example, in males, we observe increased transcription of the flamenco cluster (Fig 4G) and, in females, both the flamenco and 20A clusters (Fig 4H). This is presumably due to the presence of germ line cells where piRNA biosynthesis is constitutively active. These results strengthen and support our conclusions from the effect we observe in brains.

1.11 Discussion: authors should stress the limitations of the study. For example, autonomously generated transposon transcripts would be better quantified from 5'cap RNA-seq and not from bulk RNA-seq in which the reads can be contributed by introns (often containing TE's), also functional piRNA production can only be shown by small RNA sequencing. Possible indirect effects should also be discussed.

We appreciate the comment and recognize that for analysis of individual piRNAs, small RNA sequencing is the preferable. However, we were interested in cluster activation and therefore map reads to longer piRNA precursor sequences. We have stressed this limitation in the discussion, line 395-401:

“Functional piRNA production is best examined using small RNA sequencing, which was used by (Chaouch *et al.*, 2021) to confirm expression of piRNAs upon overexpression of H3.3K36M. We have analysed RNA-seq with paired-end read length of 150bp. Still, considering the same loci became activated upon overexpression of H3.3K36M and, in $\Delta HisC$; $12x^{H3K36R}$, as well as in $Set2^1$ and NSD^{ds46} , it seems likely that in all cases piRNAs are formed.”

Reviewer 1 – Minor points

1.12 All figure legends incorrectly spell "Volcano" plots as "Vulcano" plot .
This is corrected.

1.13 Color code in all main and EV figures should be consistent (it even changes between panel of the same figure!) and clear (Fig. 1C 2 different blues cannot be distinguished).

The colour code has been revised and is now consistent throughout the manuscript. Note that different classifications are used in different panels, e.g., Figs 1B and 1C and in these cases the same/similar colour may have different meanings between panels when different classifications are used.

1.14 *Fig 1. (A) Plot title. Replace "vs WildType" to "vs ΔHisC; 12xH3K36K" and use color code presented in 1B for TE's selected for the analysis. P<0.05 should be at 1.3 on the Y-scale (B) Expected TE distribution does not correspond to Table EV1. Example: other. Add N=4088 and N=140221 to both categories. Same is true for Fig. EV1(C) Should represent TE's upregulated according to legend: LTR/Gypsy, Pao and Copia, LINE/Jockey and other as well as DNA but not RC or other. Why is the heterochromatic part of the X chromosome that small in this representation? (D) Is "wild type" used as reference here, or Oregon R? Same question for Fig. 2A/B/C.*

We have corrected this in the titles of the plots and throughout the text. Wild type now refers only to Oregon R and $\Delta\text{HisC}; 12x^{\text{H3K36K}}$ is used in applicable cases. We have coloured the transposons and piRNAs that exceed a $|\log_2\text{FC}|$ of 2 whilst those with $p>0.05$ were filtered out. Therefore, the y-axes begin at 1.3. The information is added to figure legends, e.g., line 650-651:

"Note that the y-axis in this and all following volcano plots starts at 1.3, corresponding to $p=0.05$ "

1.15 *Fig. 2. (A/B/C) WT is Oregon R. Add color code from 1B. Y scale. (D) N=? How many of the 4088 TE's are analyzed here? If these are only upregulated TEs then why does the scale representing $\log_2\text{FC}$ ranging from -1.5 to +1.5? (E) color code consistency.*

We have revised figures 2A-C and 2E. In 2D we used repeats that were represented in at least one of the three HMT mutants and with a $\log_2\text{FC}>2$, leaving a total of 2495 TE's. The scale for 2D was chosen arbitrarily to visualize the clusters in the clearest possible way. Therefore, we have removed the scale to not cause any further confusion. We are thankful that you pointed this out.

1.16 *Fig. 3. (A/B) Number in Figure are total mis-regulated piRNA genes, rather use upregulated ones.*

Corrected.

1.17 *Table1: Mark all Tudor domain proteins, CG15930 and kots also have tudor domains.*

Corrected.

1.18 *Fig EV2. Color code consistency again. Represent the same relevant TE's as in Fig1C. Is there a difference for LINE and/or Simple repeats on the X in the K36R mutant, or is this due to an unfortunate choice of colors?*

Corrected, as pointed out, this was due to similar colours being used in the figure.

Reviewer #2

Reviewer 2 – Summary:

Lindehell et al in this manuscript discuss the role of methylation on lysine 36 on histone H3

and its requirement to control transposon activities in somatic cells. To do so authors have assessed transcript levels of transposable elements in brains dissected from third instar male larvae of Lysine to Arginine replacement Δ HisC; 12xH3K36R mutant and the reciprocal control. In agreement with Chaouch et al., 2021, authors have identified transcriptional activation of TE. Majority of which being Long Terminal Repeats (LTR) of Gypsy, Pao and Copia and Jockey family line repeats as previously seen. The genomic location of these TE are also in line with what was previously shown as the bulk being present in the pericentromeric region of all chromosomes. Authors take some thorough and interesting approach in profiling the status of TE activation in 3 of the methyltransferase mutants involved in methylation of H3 K36 and make a comparison with the K36R mutants. They find division of labor whereby each methyltransferase contributes to the process of TE activation the highest being Set2 and the other 2 less with more complementary functions. In an attempt to explain some mechanism of TE regulation authors assess TE production in different mutants and find a correlation between the increase of TE production in different methyltransferase mutants and K36R mutant. Furthermore, by analyzing chip-seq data on k36me1, me2, me3 they assess the enrichment of these marks in TE and find that highest K36me2 is found on LINE and LTR regions, suggesting me2 mark of K36 loaded by the 3 methyltransferases to be involved in the prevention of transposable element transcription. Finally, authors assess whether piRNA expression is also upregulated in Δ HisC; 12xH3K36R and the three methyltransferase mutants. They find that loss of h3 K36 methylation triggers piRNA production from dual strand piRNA clusters and further observe transcriptional activation of almost all genes linked to piRNA biogenesis.

The manuscript is very well written and it further reinforces that H3k36 is necessary for repression of TEs in somatic cells and that mutations within this lysine may cause CNS diseases and cancer as shown in literature. However, the proposed mechanism that it is through K36me2 loss is highly possible but not clearly shown/described in the manuscript and should be further clarified/explained. Special clarification should be granted when it comes to comparisons with H3K36M seq data that are done on eye discs whereas majority of transcriptomic data are performed on 3rd instar larval brains.

Reviewer 2 – Here are some points that require further explanation:

2.1 The authors show convincing data on transcriptional activation of transposable elements in absence of methylation on the replication coupled histone H3, presuming that Δ HisC; 12xH3K36R replacement loses k36 methylation mark. It will be easier to follow for the reader if this is clearly explained in the text even if it is previously published and the references are included in the manuscript. In other words, it will be good to have an overview of the me1,2,3 methylation state of this mutant that they have used right at the beginning of the manuscript.

We appreciate the comment and the information on the three methyltransferases and the consequences in the corresponding mutants is described in the introduction, lines 61-71:

"In Drosophila melanogaster, three evolutionary conserved proteins methylate H3K36. SET domain containing 2 (Set2) is the only histone methyltransferase to trimethylate H3K36 and Set2 mutants show a 10-fold reduction of H3K36me3 while bulk mono- and dimethylated H3K36 remain largely unaffected (Dorafshan et al, 2019; Larschan et al, 2007). Absent, small, or homeotic discs 1 (Ash1) can add either one or two methyl groups to H3K36 and loss-of-function in Ash1 results in a twofold reduction in H3K36me1 but no detectable loss of bulk mono- and dimethylated H3K36 (Dorafshan et al., 2019). Less is known about the substrates of Drosophila Nuclear receptor binding SET domain protein (NSD) but closely related mammalian orthologs (NSD1, NSD2 and NSD3) have been shown to mono- and dimethylate H3K36 (Li et al, 2009b; Rayasam, 2003). NSD loss-of-function mutants show no obvious changes in levels of bulk mono-, di- or trimethylated H3K36 (Dorafshan et al., 2019)."

As suggested by the reviewer we have added information on H3K36 methylation in H3K36R replacement mutant. Line 121-123 now reads:

"We have previously shown that trimethylated H3K36 is strongly reduced in brain and imaginal discs from mutant $\Delta HisC; 12x^{H3K36R}$ larvae (Lindehell et al., 2021)."

2.2 Did the authors test K36R me3, me2, me1 level in those methyltransferase mutants (set2, NSD and ash)? If not, has it been shown previously in the literature? can authors explain what is known here or what has been done?

Please see the response to comment 2.1., above.

2.3 In page 6 of the manuscript (line 162) authors ask whether the TE activation in K36R is linked to one of the methylation states of k36. They mention that they have done that by analyzing seq data for mono-, di- and trimethylated H3K36 from adult fly brains. Why is so? When all the transcriptomic data is done on third instar larval brains.

We appreciate the comment and, to our knowledge, data for all three H3K36 methylation stages have not been produced from *Drosophila* third instar larval brains. Therefore, we decided to use the most representative datasets available with all three methylation stages. We believe that the high quality modEncode data from adult heads is the most appropriate available data set to use. We have added the following sentence to motivate our use of larval brains for transcriptomic analysis, lines 105-108:

"Third instar larvae stage was chosen as it is the last viable stage for all mutant strains and, brains because it is a diploid tissue with minor maternal contribution. Males were used to address questions about possible effects of dosage compensation of the single male X (Lindehell et al., 2021)"

2.4 Related to the same question above in material and methods referring to data produced in figure 2 (E, F, EV1), authors mention page 13 (line 384) that the RNA seq is from adult heads (GSE140979) (Chaouch et al., 2021) while all chip-seq data performed in this study is done on eye discs. Can the authors explain? This also raises the question about tissue specificity of these marks as these methylation marks are measured in eye discs, therefore making the argument of the TE activation seen in larval brains of K36R due to loss of me2 not completely accurate. Additionally, when such comparison is done in

different tissues the CNS context of this observation should be reconsidered as this is tested in 2 different tissues and should be clearly stated and elaborated in text. Also, can authors explain why they chose to look into the methylation mark levels if it varies amongst different transposons in H3k36M and not the wild type H3.3 overexpression as well to have the full picture.

Please note that the replacement mutant H3.3K36M has been shown to inhibit the enzymatic activity of K36 related histone methyltransferases and thus to affect both modifications on H3.3 and H3.2. Whether the loss of the K36 methylation marks altogether and/or which function of any specific methyltransferase is responsible for observed activation of transposable elements remained elusive. This motivated our study. We conclude that all three H3K36 histone methyltransferases, presumably through the dimethylation of H3K36, help to prevent transcription of transposable elements and that Set2 is the main methyltransferase responsible for this. We agree that the link to H3K36me2 is correlative, and we have clarified this in the text. Lines 330-335 now reads:

“However, CHIP-seq data shows that H3K36me2 is significantly enriched in LINE and LTR transposable elements (Fig 2F). This result is supported by findings from polytene chromosome stainings showing H3K36me2 enrichment in pericentromeric heterochromatin (Lindehell *et al.*, 2021) which also are the genomic regions with the highest enrichment of LTR and LINE transposable elements (Fig 1C). Although correlative, this is in line with the notion that depletion of H3K36me2 in a mouse cell line largely recapitulates the transcriptional changes observed by H3.3K36M (Rajagopalan *et al.*, 2021).”

2.5 In addition I think clear distinction should be also made that sequence data of H3k36M used in the study is from overexpression studies and it might not recapitulate completely the situation with $\Delta HisC$; 12xH3K36R replacement used in this current study although it is interesting that they share somehow similar features (Fig EV1). It will be good to clearly mention this in the text.

We appreciate this comment and have made changes to the text to highlight the differences between the two conditions. Please note that in this revised version we have also added an analysis of an additional dataset, with $\Delta HisC$; 12xH3K36R replacement from whole larvae, males and females separated (Fig 4). Please see the response to comment 1.10 above.

Reviewer 2 – Minor comments

2.6 I think it should be clearly stated why male larval brains were used in the study?

The following sentence has been added to motivate our use of larval brains for the transcriptomic analysis, lines 105-108:

“Third instar larvae stage was chosen as it is the last viable stage for all mutant strains and, brains because it is a diploid tissue with minor maternal contribution. Males were used to address questions about possible effects of dosage compensation of the single male X (Lindehell *et al.*, 2021)”

2.7 Page 11 Discussion (line 326, it seems unlikely to line 329 same effect) is unclear and requires some sentence breakdown and more explanation.

We appreciate the comment and have revised this passage which now reads:

“It is thought that H3K36R replacement inhibits PRC2-mediated methylation and may be structurally analogous to inhibition of PRC2 by di- or trimethylated H3K36 (Finogenova et al., 2020; Jani et al., 2019). It seems unlikely, that the increased transcription of transposable elements in H3K36R larvae is caused by impaired PRC2 activity. This would require that the loss of H3K36 methyltransferases (the loss of PRC2 inhibitors) and the H3K36R replacement (PRC2 inhibition) lead to the same effect.”

2.8 I recommend moving Fig EV3 from supplement to main as it will be nice to see the whole picture of piRNA expression in all three methyl transferases.

We agree and as suggested we have moved the panels previously in Fig EV3 to main Figure 2.

Reviewer #3

The manuscript 'Methylation of lysine 36 on histone H3 is required to control transposon activities in somatic cells' dissections the effect of three histone H3 K36 methyltransferases, Set2, NSD and Ash1, in the eye of Drosophila melanogaster. As previously described for loss of H3K36 methylation in Drosophila eyes, the authors observe an upregulation of transposon transcripts (Figure 1). Figure 2 shows that the upregulation of TE transcripts is strongest in knock-down of Set2, the only MTase that tri-methylates H3K36. Knockdown of NSD and Ash1, which deposit mono- and di-methylation on H3K36, show minor effects on TE transcript abundance. The authors observe increased transcript abundance for mostly LTR-derived sequences in Set2 knock-down (Fig. 2E). However, such LTR elements do not show enrichment for H3K36me3, which is deposited by Set2 (Fig. 2F). Both LTR and Non-LTR (LINE) elements show enrichment of H3K36me2 by ChIP. In Fig. 2F enrichment of H3K36me2 and me3 is mostly observed across exons.

The authors then start talking about PIWI-interacting RNAs (piRNAs). Experimentally they seem to have compared short RNA-seq reads, which technically represent parts of longer RNAs (>200nt for standard RNAseq protocols), to sequences in a piRNA-database. The authors wrongly call their analysis 'piRNA analysis'. Figure 3 and the corresponding parts of the manuscript and method exhibit a complete lack of understanding of what piRNAs are: PIWI interacting RNAs (piRNAs) are small RNAs that are defined by their physical association with PIWI proteins (Girard et al., 2006). To interrogate piRNAs, the authors must (1) show the presence of one of the three fly PIWI proteins, which are germline-restricted genes under physiological conditions, (2) immunoprecipitated the expressed PIWI protein, and (3) prepare the associated small RNAs for next-generation sequencing. I recommend the authors to familiarize themselves with piRNA biology: Reviews and primary literature from the Hannon, Zamore, Brennecke, Aravin and Siomi and other labs.

This manuscript seems to be based on a poor understanding of transposons and a complete misunderstanding of piRNA biology. The tested hypotheses, experimental approaches and interpretations are fundamentally flawed.

We acknowledge that the reviewer comments on our use of “piRNA” as a term. This is a valid critique that we value as a point to improve and, we have now corrected this mistake. As pointed out by the reviewer we measure transcripts originating from piRNA clusters, i.e., putative piRNA precursors, the prerequisite for piRNA production. In this revised version we have throughout the text corrected this usage of terms and instead of “piRNA” we use the more stringent “transcription from piRNA clusters”. We recognize that for analysis of individual piRNAs, small RNA sequencing is the preferable. We have stressed this limitation in the discussion, line 395-401:

“Functional piRNA production is best examined using small RNA sequencing, which was used by (Chaouch *et al.*, 2021) to confirm expression of piRNAs upon overexpression of H3.3K36M. We have analysed RNA-seq with paired-end read length of 150bp. Still, considering the same loci became activated upon overexpression of H3.3K36M and, in $\Delta HisC$; $12x^{H3K36R}$, as well as in $Set2^1$ and NSD^{ds46} , it seems likely that in all cases piRNAs are formed.”

We do acknowledge the critique, still, the comment of this reviewer is very passionate, condescending, and offensive and not in line with “*Referees are asked to maintain a positive and impartial, but critical, attitude in evaluating manuscripts. Criticisms should remain dispassionate; offensive language is not acceptable.*”

Reviewer 3 – Major points

3.1 Fig. 1 The main message has already been described in Chaouch et al., 2021.

We respectfully disagree to this comment. Please note that the replacement mutant H3.3K36M has been shown to inhibit the enzymatic activity of K36 related histone methyltransferases and thus to affect both modifications on H3.3. and H3.2. Whether the loss of the K36 methylation marks altogether and/or which function of any specific methyltransferase is responsible for observed activation of transposable elements remained elusive. The latter is important since two out of the three K36 related methyltransferases may have non-histone targets in addition to H3K36 which have severe effects on gene-regulatory functions (Dorafshan et al., 2019; Lindehell et al., 2021). These questions and concerns motivated our study.

Reviewer 3 – Minor issues

3.2 The authors observe changes in steady state RNA levels not activation of transcription. The title of this figure should be rephrased accordingly.

We appreciate this comment and have revised throughout the manuscript accordingly. We have also added the following sentence to clarify, lines 126-127:

“In our analysis, we measure increased transcript abundance which we assume is mainly caused by increased transcription.”

3.3 Fig. 1C: This panel indicates the genomic location of all possible TE insertions that could be responsible for the observed increase in TE transcript levels. Due to multimappability of TE-derived reads in RNA-seq, the authors cannot for sure determine their genomic origin. The interpretation of this depiction should be rephrased accordingly or remove this figure panel.

Please see also response to comments 1.3. and 1.7. above. We have added the following statement to the discussion to clarify the multimapping problem when dealing with repeats, line 366-367:

“We note that due to multimapping of reads we cannot determine the exact origin of transcribed transposable elements.”

3.4 Fig. 2. is the first figure that contains novelty and could be merged with Fig 1.

We have discussed this comment. With the addition of panel 1D we feel that figures 1 and 2 should be kept as two separate figures.

3.5 Fig. 2F shows enrichment of H3K36me2 and m3 mostly on 'exons'. Are these protein coding exons or all exons? A differential expression of these regions should be shown, and they might be the most prominent changes of the transcriptome.

We have added Figure 1D which shows that transcribed transposons primarily originate from intergenic regions and if located within genes, these are genes which remain silent in H3K36R, Figure 1D.

3.6 Figure 3. The presented 'piRNA' data are an artifact based on a complete misunderstanding of the fundamental definition of piRNAs. I recommend the authors to start familiarizing themselves with the basic literature on piRNA biology.

We acknowledge the reviewer comment, we have corrected the terminology used and have done as suggested.

May 1, 2023

RE: Life Science Alliance Manuscript #LSA-2022-01832-TR

Prof. Jan Larsson
Umeå University
Molecular Biology
Dept of Molecular Biology
Umeå SE90187
Sweden

Dear Dr. Larsson,

Thank you for submitting your revised manuscript entitled "Methylation of lysine 36 on histone H3 is required to control transposon activities in somatic cells". We would be happy to publish your paper in Life Science Alliance pending final revisions necessary to meet our formatting guidelines.

-please add the author contributions and a conflict of interest statement to the main manuscript text

A. FINAL FILES:

B. MANUSCRIPT ORGANIZATION AND FORMATTING:

****It is Life Science Alliance policy that if requested, original data images must be made available to the editors. Failure to provide original images upon request will result in unavoidable delays in publication. Please ensure that you have access to all original**

data images prior to final submission.**

The license to publish form must be signed before your manuscript can be sent to production. A link to the electronic license to publish form will be sent to the corresponding author only. Please take a moment to check your funder requirements.

Sincerely,

Reviewer #1 (Comments to the Authors (Required)):

In the revised version of the manuscript entitled "Methylation of lysine 36 on histone H3 is required to control transposon activities in somatic cells" the authors show that upon replacement of the replication-dependent histone H3 by an H3K36R mutant, transposable elements (TE's) are derepressed in somatic cells of the brain. This de-repression is correlated with an increased expression of transcripts from the piRNA cluster suggesting that this pathway can also be activated at least in part in non-germline somatic cells. The authors show that the three HMTs which are known methylate K36 may share the labor to maintain transposable elements repressed with Set2 playing a major role.

The authors have improved the quality of their manuscripts according to reviewer comments.

I appreciate in particular:

- the parralled analysis of an independent data set (Meers, 2017) which is presented in fig. 4. and mostly confirms the authors results. It also highlights that a significant number of TE's may also end up repressed rather than de-repressed. Because this is also observed in particular for NSD and Ash1 mutants, the author's should at least mention this difference and discuss it.

Note: Fig.4 however may be presented in EV to keep the short format recommended for LSA publications.

- Mentions of the limitations of the study concerning the methylation state needed for repression of TE's, the impossibility to clearly state the genome-wide localization of derepressed copies of the TE affected in each family and also the impossibility to claim bona fide piRNA may be generated.

Altogether, the authors addressed all my main concerns, I support the publication of their manuscripts with minor revisions in LSA.

Reviewer #2 (Comments to the Authors (Required)):

Lindehell et al in this manuscript discuss the role of methylation on lysine 36 on histone H3 and its requirement to control transposon activities in somatic cells. The authors have responded to all my comments appropriately and modified the manuscript accordingly.

May 3, 2023

RE: Life Science Alliance Manuscript #LSA-2022-01832-TRR

Prof. Jan Larsson
Umeå University
Molecular Biology
Dept of Molecular Biology
Umeå SE90187
Sweden

Dear Dr. Larsson,

Thank you for submitting your Research Article entitled "Methylation of lysine 36 on histone H3 is required to control transposon activities in somatic cells". It is a pleasure to let you know that your manuscript is now accepted for publication in Life Science Alliance. Congratulations on this interesting work.

DISTRIBUTION OF MATERIALS:

Again, congratulations on a very nice paper. I hope you found the review process to be constructive and are pleased with how the manuscript was handled editorially. We look forward to future exciting submissions from your lab.

Sincerely,
